# Actial: Activate Spatial Reasoning Ability of Multimodal Large Language Models

**Xiaoyu Zhan**[1*], **Wenxuan Huang**[3,4*†], **Hao Sun**[1*], **Xinyu Fu**[1], **Changfeng Ma**[1]
**Shaosheng Cao**[2✉], **Bohan Jia**[3], **Shaohui Lin**[3], **Zhenfei Yin**[6], **Lei Bai**[5],
**Wanli Ouyang**[4], **Yuanqi Li**[1], **Jie Guo**[1], **Yanwen Guo**[1✉]

[1] Nanjing University
[2] Xiaohongshu Inc.
[3] East China Normal University
[4] The Chinese University of Hong Kong
[5] Shanghai Jiao Tong University
[6] University of Oxford

{zhanxy, hao.sun, xinyu.fu, changfengma}@smail.nju.edu.cn
osilly0616@gmail.com, caoshaosheng@xiaohongshu.com
{yuanqili, guojie, ywguo}@nju.edu.cn

## Abstract

Recent advances in Multimodal Large Language Models (MLLMs) have significantly improved 2D visual understanding, prompting interest in their application to complex 3D reasoning tasks. However, it remains unclear whether these models can effectively capture the detailed spatial information required for robust real-world performance, especially cross-view consistency, a key requirement for accurate 3D reasoning. Considering this issue, we introduce Viewpoint Learning, a task designed to evaluate and improve the spatial reasoning capabilities of MLLMs. We present the Viewpoint-100K dataset, consisting of 100K object-centric image pairs with diverse viewpoints and corresponding question-answer pairs. Our approach employs a two-stage fine-tuning strategy: first, foundational knowledge is injected to the baseline MLLM via Supervised Fine-Tuning (SFT) on Viewpoint-100K, resulting in significant improvements across multiple tasks; second, generalization is enhanced through Reinforcement Learning using the Group Relative Policy Optimization (GRPO) algorithm on a broader set of questions. Additionally, we introduce a hybrid cold-start initialization method designed to simultaneously learn viewpoint representations and maintain coherent reasoning thinking. Experimental results show that our approach significantly activates the spatial reasoning ability of MLLM, improving performance on both in-domain and out-of-domain reasoning tasks. Our findings highlight the value of developing foundational spatial skills in MLLMs, supporting future progress in robotics, autonomous systems, and 3D scene understanding.

## 1 Introduction

Multimodal Large Language Models (MLLMs) [17, 28, 36, 39, 1, 7, 8, 42] have recently achieved significant advances in visual understanding and inference. Naturally, the researchers [34, 33, 30, 45] demonstrate considerable interest in their abilities for spatial reasoning tasks.

---

* : Equal contribution.   † : Project leader.   ✉ : Corresponding authors.

39th Conference on Neural Information Processing Systems (NeurIPS 2025).

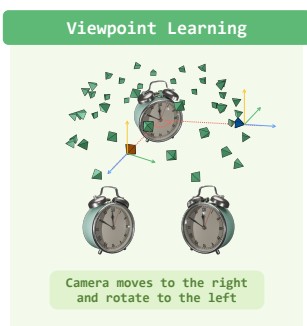
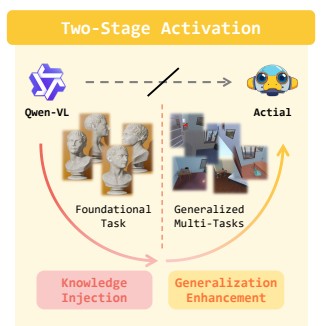
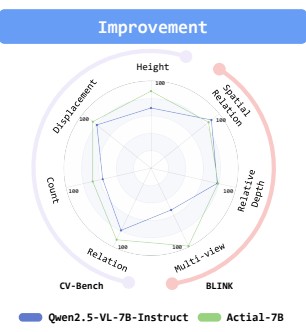

Figure 1: We aim to activate the MLLM's spatial reasoning ability with Viewpoint Learning and the two-stage fine-tuning strategy.

In the computer vision field, addressing 3D tasks typically requires first establishing cross-view consistency through methods such as camera calibration [40] and stereo matching [4, 2]. However, recent studies [29, 30] aim to enable MLLMs to directly perceive such consistency from multi-view images or sequential video frames, allowing accurate spatio-temporal reasoning. This development raises a critical question: Do these MLLMs have the potential to capture the fine-grained 3D spatial information needed for robust and reliable visual-spatial performance in real-world 3D scenarios?

As a rule, all 3D objects maintain 3D consistency in space, characterized by stable spatial relationships and geometric properties. This consistency ensures that objects retain 2D continuity in timeline when projected onto a 2D plane. Our key concern is whether existing MLLMs, which are trained primarily on video data emphasizing 2D continuity, can achieve an understanding of spatial 3D consistency, as opposed to merely tracking continuous pixel-level evolution or correlated pixel mappings. Furthermore, the camera projection commonly introduces subtle distortions, imperceptible to humans, which complicate the establishment of relationships between 2D continuity and 3D consistency.

Previous work [45, 21, 48] has made it evident that current MLLMs still struggle to capture cross-view consistency. However, additional spatial information [21, 9] and visual prompts [22, 19] can effectively improve their spatial reasoning ability. Although MLLMs do not seem to have acquired an explicit understanding of fundamental 3D mapping relationships, they demonstrate sensitivity to simple prompts that implicitly relate to these spatial concepts. We believe that the visual-spatial intelligence [45] of MLLMs has not been fully exploited due to inappropriate data utilization. Exploring how to effectively leverage existing data to teach MLLMs to reason and solve problems in 3D space represents a highly valuable research direction.

In response to this challenge, we focus on a fundamental yet essential task, **Viewpoint Learning**, aimed at evaluating and activating the spatial reasoning ability in MLLM. Identifying viewpoints in image pairs or videos constitutes a pivotal step towards achieving an understanding of 3D consistency. This task is advantageous due to its straightforward data acquisition process, ease of ground-truth calculation, and simple evaluation metrics. Capitalizing on these benefits, we introduce the Viewpoint-100K dataset, which comprises 100K real-world, object-centric image pairs captured from distinct viewpoints, each paired with ego-centric or object-centric question-answer pairs (QAs).

To effectively activate the spatial reasoning ability of MLLMs, we propose a two-stage fine-tuning strategy. The first stage is dedicated to the injection of foundational knowledge, emphasizing the critical importance of viewpoint understanding in both video comprehension and spatial reasoning. For this purpose, we use Supervised Fine-Tuning (SFT) with our Viewpoint-100K dataset, which ensures that the model develops a correct understanding of spatial relationships and viewpoint transformations. To maintain the coherent reasoning process and instruction-following behavior, we additionally employ a hybrid cold-start initialization enhanced by human-assisted pseudo chain-of-thoughts (CoTs). In the second stage, our aim is to preserve the acquired viewpoint-related knowledge while simultaneously improving the model's generalization capacity across broader spatial tasks. We apply Reinforcement Learning (RL) through the Group Relative Policy Optimization (GRPO) algorithm [32], further fine-tuning the model on the SAT dataset [30], a synthetic dataset for spatial

aptitude training. This phase is designed to refine the model's ability to transfer knowledge from basic viewpoint tasks to more abstract and complex spatial reasoning challenges. It enables models to better perceive, interpret, and reason about 3D space, which are critical skills for deployment in real-world applications requiring advanced spatial reasoning ability.

Our experiments across multiple benchmarks demonstrate the effectiveness of Viewpoint Learning in activating the spatial reasoning ability in MLLMs. In particular, this enhancement extends to out-of-domain inference tasks, showcasing the versatility and robustness of models trained with our approach. As MLLMs continue to evolve, fundamental tasks like Viewpoint Learning will play a crucial role in advancing their ability to understand and interact with the world in three dimensions, paving the way for more sophisticated applications in autonomous systems, robotics, and beyond.

In summary, the main contributions of this work are threefold.

- We introduce viewpoint learning, by which can activate the spatial reasoning ability in MLLMs, leading to strong out-of-domain generalization capabilities in visual and spatial reasoning.
- We propose the Viewpoint-100K dataset, including 100K auto-generated ego-centric or object-centric QAs based on real-world, object-centric image pairs.
- We employ a two-stage fine-tuning strategy that involves foundational knowledge injection and generalization enhancement, aiming to effectively achieve viewpoint learning. We further present a hybrid cold-start initialization method to maintain the reasoning thinking.

## 2 Related Work

### 2.1 Multimodal Large Language Models for Spatial Reasoning.

MLLMs [17, 28, 36, 39, 1, 7, 8, 42] have demonstrated exceptional capability across various tasks. Recently, several studies [21, 14, 9, 27, 5, 33, 34, 29, 30, 49, 15, 41] have been devoted to applying MLLMs to the field of 3D Reasoning.

[21] introduces coarse, object-level correspondences into the input images, enhancing the spatio-temporal reasoning capabilities of MLLMs without the need for fine-tuning. This work reveals that current MLLMs struggle to capture cross-view consistency but that this limitation can be mitigated by providing additional cross-view information. Several recent studies [14, 9, 27, 5] have similarly advanced the integration of MLLMs with 3D environments by incorporating rich 3D inputs and features, further demonstrating the potential of grounding language models in spatial contexts. Spatial-MLLM [41] aims to improve spatial understanding through high-quality, diverse multi-task data and additional 3D features introduced via a VGGT [38] backbone. MLLM-for3D [15] treats 3D consistency as an external prior to align 2D reasoning results across views in 3D space and gains significant improvements in spatial tasks. In a related direction, SpatialMM [33] shows that the inclusion of bounding boxes and scene graphs significantly improves spatial reasoning performance, particularly for tasks that involve fewer reasoning steps.

Based on these findings, recent studies have increasingly emphasized the need to develop targeted training strategies that address specific deficiencies in visual-spatial reasoning. However, many existing work focus on high-level reasoning capabilities, they often overlook the importance of foundational tasks for spatial reasoning such as viewpoint estimation and spatial transformation.

### 2.2 Benchmarks for Spatial Reasoning

Benchmarks play a crucial role in evaluating and advancing models' capabilities in spatial reasoning. Various datasets [10, 48, 23, 30, 45, 18, 34, 12, 20, 26, 44, 35] have been developed to assess different aspects of spatial understanding, such as identifying reference frames [48, 23, 25], handling multi-view data [30, 45], and interpreting complex scene graphs [34, 43, 26].

BLINK [12] and 3DSRBench [25] focus on various perception and reasoning tasks. SAT [30] addresses spatial reasoning through a procedurally generated, multi-task dataset built on synthetic data. It also identifies the limitations of MLLM in handling camera movement and out-of-domain relations. VSR [23] highlights the importance of identifying reference frames in spatial reasoning, showing that this capability significantly enhances the accuracy and contextual awareness of 3D environment interpretations. COMFORT [48] further explores how MLLMs respond to different

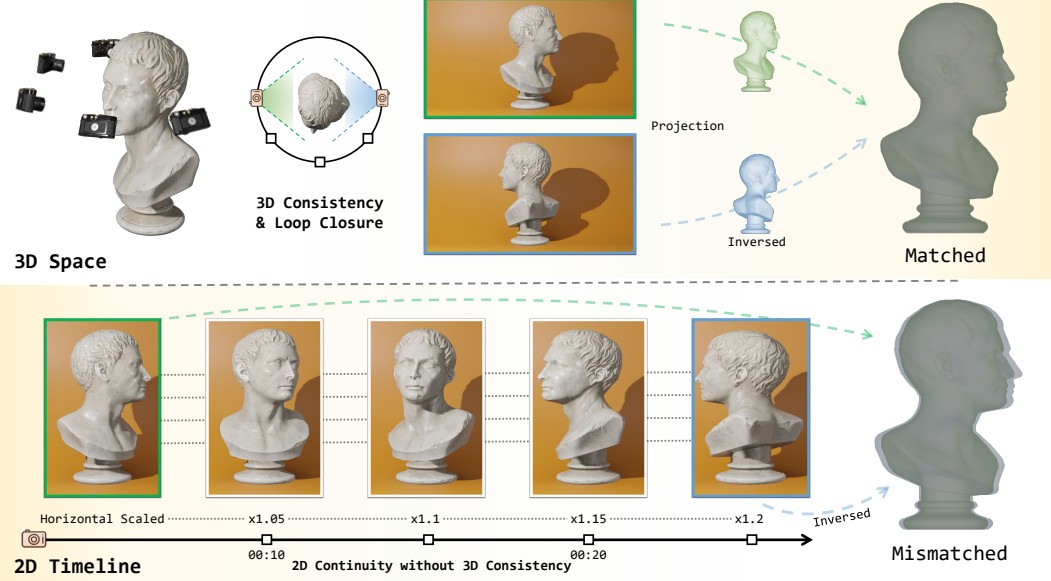

Figure 2: **2D Continuity and 3D Consistency.** 2D continuity refers to the high similarity between adjacent frames, whereas 3D consistency focuses on preserving spatial and geometric relationships across frames. **Top**: Verifying 3D consistency requires estimating the camera pose and comparing these spatial properties in 3D space. **Bottom**: Adjusting the scale of each video frame slightly can destroy 3D consistency while maintaining 2D continuity.

frames of reference, revealing their sensitivity to such variations and a tendency to favor English-specific conventions when resolving spatial ambiguities. VSI-Bench [45] presents eight tasks in three categories to assess visual-spatial intelligence. It indicates that most errors stem from spatial reasoning challenges, particularly relational reasoning mistakes and difficulties in transforming between egocentric and allocentric perspectives. The study also finds that linguistic prompting techniques can be detrimental to spatial reasoning performance.

Together, these works highlight the strong potential of MLLMs in understanding and reasoning about visual-spatial information. In this paper, we aim to investigate how such spatial reasoning ability can be effectively activated and leveraged in MLLMs for complex visual and spatial reasoning tasks.

# 3 Overview

For the successful execution of spatial tasks using 2D data representations such as images and videos, it is essential to recognize and leverage the inherent 3D consistency of objects. While 2D continuity in videos focuses on seamless transitions between frames through subtle changes that ensure a smooth visual experience, true 3D consistency requires preserving spatial integrity and geometric relationships across frames, including depth, scale, and object positions. This makes 3D consistency inherently more complex than 2D continuity. Although 3D consistency can be maintained after projecting to 2D planes, achieving 2D continuity alone does not guarantee 3D consistency (Figure. 2). This distinction is crucial for tasks such as 3D reconstruction, SLAM, and pose estimation.

In recent progress in video generation, models [24, 3, 16] still struggle to replicate 3D properties in the real world, such as perspective consistency and vanishing points. Interestingly, humans, despite living in a 3D world, often fail to detect such inconsistencies in 2D sequences, revealing the subtlety of spatial perception. MLLMs, typically trained on 2D data and constrained by low frame rates due to memory limits, face significant challenges in achieving reliable 3D reasoning without explicit spatial supervision or enriched multimodal inputs.

We argue that activating MLLMs' spatial reasoning ability hinges on correcting their conceptual understanding of visual input, specifically the ability to recognize and leverage the 3D consistency of objects. Images and videos should not be seen merely as sequences of changing pixels but as

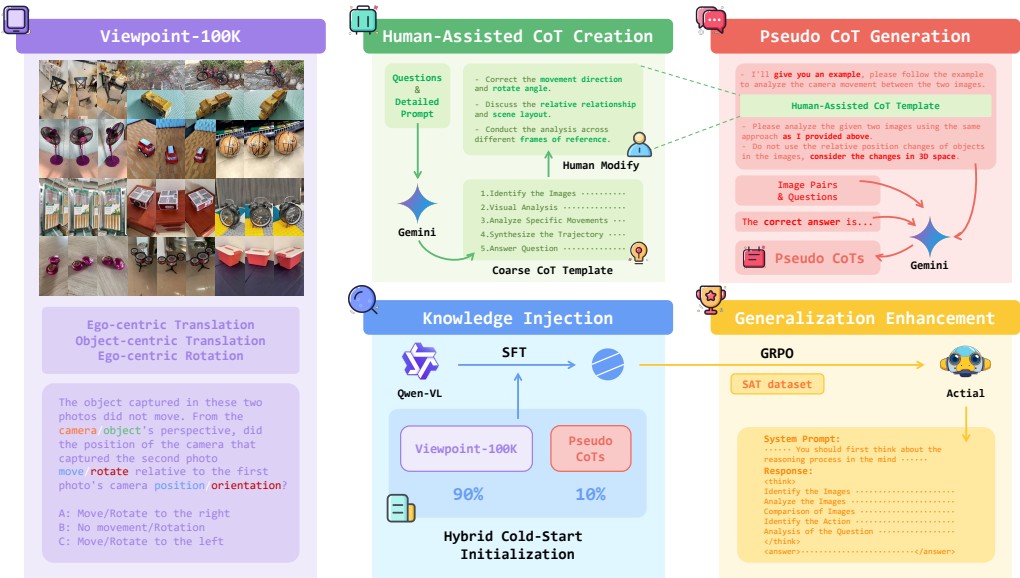

Figure 3: **Overview of our pipeline.** We introduce Actial, which comprises a novel dataset and a two-stage fine-tuning strategy. In the knowledge injection phase, we employ a hybrid cold-start initialization to enhance the model's foundational spatial skills and leverage pseudo CoTs to ensure robust reasoning capabilities. Subsequently, we enhance the model's generalization capabilities through a specialized generalization enhancement stage.

continuous projections of 3D space onto a 2D plane. Recognizing that 2D continuity is supported by 3D consistency enables models to better interpret and reason about spatial structure.

## 4 Method

Given the strong generalization capabilities of MLLMs, the key challenge is enabling them to grasp the structure of 3D space. Our goal is to make these models realize that multi-view images and videos are not merely sequences of 2D representations but rather projections of 3D-consistent objects onto a 2D plane. As shown in Figure. 3, we introduce Viewpoint Learning in the Section. 4.1, Foundational Knowledge Injection in the Section. 4.2, Hybrid Cold-Start Initialization in the Section. 4.3 and Generalization Enhancement in the Section. 4.4.

### 4.1 Viewpoint Learning

To teach MLLMs how to handle 3D visual tasks, it is essential to help them perceive 3D consistency. Specifically, 3D consistency ensures that objects in 3D space maintain 2D continuity when projected onto a 2D plane. This property enables us to perform 3D reasoning based on the correlations among 2D images captured from different viewpoints. To recognize this consistency, MLLMs must first understand the concept of viewpoints.

It is hard to ask MLLMs to directly regress accurate camera poses from multi-view images. To make the question easier for the MLLMs, we decide to simplify the problem. Considering that the movement of the camera in space can be decomposed into two stages: translation and rotation. We will separate these two problems and abstract them into simpler multiple-choice questions, rather than precise regression problems.

**Question Setting.** The challenges of top-down and left-right relative positioning are fundamentally similar. As most images in MVImgNet are captured through horizontal loop shooting, we limit our question generation to horizontal transformations (horizontal translation and rotation).

Inspired by [18, 48], which propose the importance of frames of reference (FoR). We generate mainly three types of question. They are ego-centric camera translation and rotation centered around the

camera's perspective and object-centric camera translation centered around the object's perspective. Examine the two types of thinking, translation and rotation, of MLLMs in viewpoint cognition, as well as their spatial perception ability in two reference frames (ego-centric and object-centric).

**Data Generation.** We automatically generate Viewpoint-100K from MVImgNet [47], including 100K object-centric image pairs and the corresponding QAs. MVImgNet is a large-scale multi-view image dataset containing approximately 6.5 million real-captured frames, along with precise camera calibrations. For each subject, the dataset provides a comprehensive set of object-centric images, including corresponding object masks, camera intrinsic and extrinsic parameters, depth maps, and point clouds.

For each sample in the Viewpoint-100K dataset, we generate image pairs by randomly selecting two images of the same subject with different viewpoints from MVImgNet. We ensure that the horizontal angle between the camera viewpoints is between 20 and 100 degrees. Using the provided camera parameters, we compute the relative translation and rotation between the two views. As specified in the problem setup, we only consider the camera translation along the horizontal axis and its rotation around its own vertical axis. The final dataset encompasses a total of 10,813 distinct objects, which belong to 205 different object classes.

## 4.2 Foundational Knowledge Injection

When evaluating MLLMs on the Viewpoint-100K dataset, we observed that baseline models primarily depend on 2D visual cues for viewpoint-related tasks (shown in the Figure. 4), resulting in accuracy levels near random guessing. This finding indicates that these models do not take advantage of 3D consistency for spatial reasoning across multiple views. Our goal is to enhance these models so that they can effectively utilize 3D spatial features rather than rely solely on superficial 2D features. This shift from 2D cues to 3D consistency is essential for achieving robust performance in complex, multi-view spatial reasoning tasks.

Inspired by prior research [13, 32], we initially employed reward-based fine-tuning using the Group Relative Policy Optimization (GRPO) algorithm to guide the model towards more sophisticated 3D spatial reasoning. Despite these efforts, we observed consistently high KL divergence during training, indicating a significant departure from the initial policy and suggesting an entrenched bias towards 2D reasoning inherent in the pre-trained models. This outcome reveals that straightforward reinforcement learning strategies are inadequate for overcoming the strong inductive biases acquired during large-scale pre-training. This emphasizes the necessity for adopting more targeted methods to foster effective 3D spatial reasoning in MLLMs.

We find that directly applying Supervised Fine-Tuning (SFT) on the Viewpoint-100K dataset leads to a substantial improvement in the model's spatial reasoning ability. Since viewpoint understanding is a core aspect of 3D perception and a direct indicator of spatial reasoning ability, explicit supervision on this task helps mitigate the 2D-centric biases acquired during pre-training. Our experiments demonstrate that SFT-based training on Viewpoint-100K not only strengthens the model's spatial reasoning capabilities but also reduces its dependence on superficial 2D visual cues, fostering a more structured and accurate understanding of 3D spatial relationships.

## 4.3 Hybrid Cold-Start Initialization

We note that the instruction-following capability and the thinking process of the model is affected after SFT. Drawing inspiration from [13], we propose to address this issue by introducing the cold-start initialization. To protect the integrity of the injected knowledge, we present to use the hybrid cold-start initialization. This means combining the CoT templates with the Viewpoint-100K dataset as the input for SFT, make the model can simultaneously learn viewpoint representations and maintain coherent reasoning thinking.

Directly having MLLMs generate CoT based on the problem often leads to incorrect thinking patterns. As an alternative, we manually constructed a correct CoT template based on MLLM's raw output. Then we provide the template, images, questions, and corresponding correct answers to MLLM, allowing it to analyze the images based on the template and attempt to answer the questions with reference to the correct answers we provide. Specifically, we use Gemini 2.5 Pro [36] to generate 1K pseudo CoTs upon Viewpoint-100K. It should be noted that there are still some analyses in the

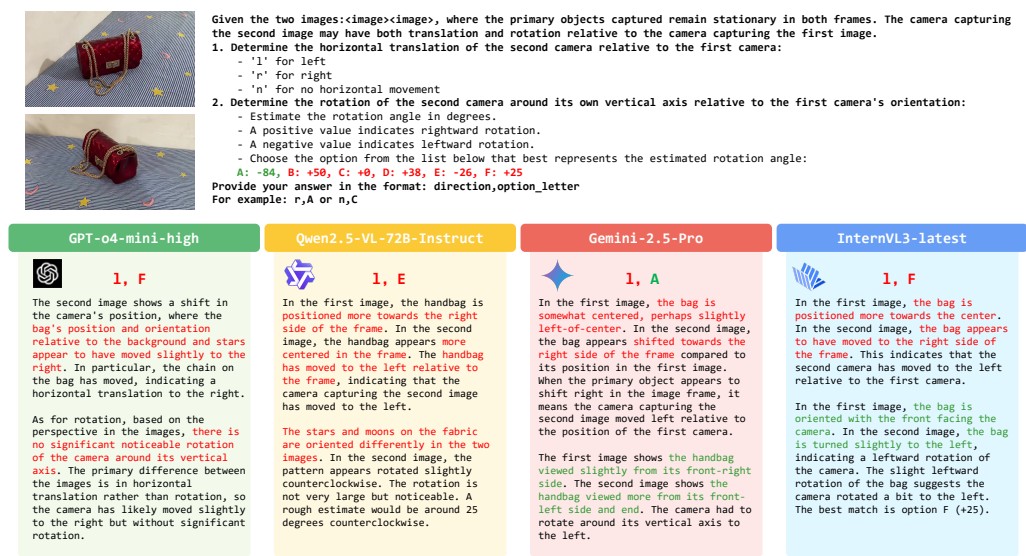

Figure 4: **Thoughts on viewpoint question.** Current MLLMs tend to rely on 2D cues to address viewpoint-related problems, which often leads to incorrect reasoning and erroneous results.

pseudo CoTs that contradict correct 3D knowledge or do not match the input image. Therefore, we maintain a relatively small proportion of pseudo CoTs in the hybrid cold-start initialization to avoid disrupting the model's learning of accurate viewpoint knowledge.

### 4.4 Generalization Enhancement

Given the critical role of viewpoint tasks in achieving cross-view consistency, we observed significant performance improvements across multiple benchmarks after incorporating foundational knowledge. However, focusing solely on a single viewpoint task during fine-tuning risks overfitting, which can limit the model's effectiveness in broader perception and inference tasks. To cover this issue, we propose a second phase of fine-tuning on a more diverse dataset. This additional phase aims to broaden the model's capabilities, ensuring it can perform robustly across a wider range of tasks and scenarios. By extending fine-tuning to a richer and more varied dataset, we expect to achieve even greater performance enhancements and improve the model's adaptability for complex real-world applications.

We select the SAT dataset [30] for our generalization enhancement phase. SAT is a synthetic spatial aptitude training dataset designed to evaluate both static and dynamic spatial reasoning. We employ Reinforcement Learning, specifically the Group Relative Policy Optimization (GRPO) algorithm. Reward-based optimization encourages the model to generate its own reasoning chains and apply the spatial knowledge acquired earlier, fostering deeper and more flexible understanding. By avoiding direct supervision on intermediate reasoning steps, our approach ensures that the model retains its previous knowledge about viewpoints and effectively leverages newly acquired spatial reasoning capabilities. This enhancement significantly improves the model's adaptability and robustness, facilitating superior performance on both in-domain and out-of-domain tasks. Furthermore, it enables the model to effectively accommodate a broader range of datasets.

## 5 Experiments

### 5.1 Datasets and Details

**Training datasets.** Actial performs the two-stage fine-tuning strategy. We use Viewpoint-100K training set for knowledge injection and SAT training set [30] for generalization enhancement.

Table 1: **Evaluation on 3DSRBench [25], CV-Bench [37], and BLINK [12].** We color the `best`, `second-best`, and `third-best` results. We also color the `better` ablation results. ↑ indicates improvement over the baseline. K.I. means knowledge injection (SFT phase), G.E. means generalization enhancement (GRPO phase).

| Model | 3DSRBench | | | | | CV-Bench | | | | | BLINK | | | |
|---|---|---|---|---|---|---|---|---|---|---|---|---|---|---|
| | Avg. | Height | Loc. | Orient. | Multi. | Avg. | Rel. | Count | Disp. | Depth | Avg. | MultiView | RelDep | SpRel |
| *Chance* | | | | | | | | | | | | | | |
| Chance Level (Random) | - | - | - | - | - | - | 50.0 | 22.5 | 50.0 | 50.0 | - | 50.0 | 50.0 | 50.0 |
| *Proprietary Models* | | | | | | | | | | | | | | |
| GPT-4o | 44.6 | 51.6 | 60.1 | 21.4 | 40.2 | 79.4 | 85.7 | 65.9 | 78.2 | 87.8 | 62.7 | 55.6 | 59.7 | 72.7 |
| Gemini-1.5-Flash | - | - | - | - | - | 71.6 | 76.9 | 66.0 | 68.3 | 75.3 | 59.0 | 51.1 | 62.9 | 62.9 |
| Gemini-1.5-Pro | 50.3 | 52.5 | 65 | 36.2 | 43.3 | 77.7 | 85.2 | 70.4 | 72.8 | 82.4 | 59.2 | 36.8 | 70.2 | 70.6 |
| Gemini-2.0-Flash | 49.8 | 49.7 | 68.9 | 32.2 | 41.5 | - | - | - | - | - | - | - | - | - |
| QwenVLMax | 52.4 | 45.5 | 70.5 | 39.7 | 44.8 | - | - | - | - | - | 71.1 | 40.6 | 84.7 | 88.1 |
| *Open-Source Models* | | | | | | | | | | | | | | |
| Robopoint-13B | - | - | - | - | - | 69.7 | 79.4 | 53.6 | 71.3 | 74.7 | 58.4 | 48.1 | 51.6 | 75.5 |
| LLaVA-v1.5-7B | 38.1 | 39.1 | 46.9 | 28.7 | 34.7 | - | - | - | - | - | - | - | - | - |
| LLaVA-v1.5-13B (+SAT) | - | - | - | - | - | 76.2 | 89.7 | 61.5 | 73.0 | 80.7 | 64.6 | 44.4 | 76.6 | 72.7 |
| LLaVA-Vid-7B (+SAT) | - | - | - | - | - | 78.7 | 81.2 | 66.2 | 79.3 | 88.2 | 62.6 | 48.1 | 66.1 | 73.4 |
| Cambrian-8B | 42.2 | 23.2 | 53.9 | 35.9 | 41.9 | - | - | - | - | - | - | - | - | - |
| *Baseline* | | | | | | | | | | | | | | |
| Qwen-2.5-VL-7B-Instruct | 45.8 | 42.8 | 59.3 | 39.3 | 38.8 | 71.2 | 79.3 | 56.9 | 79.6 | 69.1 | 73.4 | 53.3 | 78.2 | 88.8 |
| **Actial-7B** (Ours) | 47.7↑ | 46.4↑ | 60.3↑ | 35.5 | 43.0↑ | 83.5↑ | 91.2↑ | 68.7↑ | 85.6↑ | 88.5↑ | 87.6↑ | 99.2↑ | 79.0↑ | 84.6 |
| - w/o K.I. | 47.6 | 44.9 | 57.2 | 39.4 | 44.1 | 83.1 | 90.3 | 71.5 | 84.8 | 86.1 | 74.9 | 55.6 | 83.8 | 85.3 |
| - w/o G.E. | 46.6 | 47.1 | 63.3 | 27.9 | 41.2 | 73.1 | 88.6 | 61.5 | 59.0 | 83.5 | 86.6 | 99.2 | 79.0 | 81.8 |

**Evaluation benchmarks.** We use 3DSRBench [25], CV-Bench [37], and BLINK [12] to evaluate the model's abilities for spatial reasoning.

**Training details.** We use Qwen2.5-VL-7B-Instruct [1] as our baseline model. In the SFT phase, we trained for 2 epochs with a learning rate of 5e-6, a batch size of 128 and 50 warm-up steps. We mix the Viewpoint-100K dataset and pseudo CoT data as inputs. The interleave ratio is set to 0.9:0.1.

In the GRPO phase, we trained for 150 steps with a learning rate of 1e-6 and a batch size of 1024. The model is trained from post-SFT model with an 4K token generation limit, sampling 16 samples per input. During training, we set the Kullback–Leibler (KL) penalty [32, 31] to 0.2 and 1e-2 for the hyper-parameters $\epsilon$ and $\beta$, respectively. Within the reward function, the format reward and the result reward are each assigned a score of 0.5.

## 5.2 Evaluations

We evaluate Actial across multiple benchmarks using VLMEvalKit [11]. The performance results of various MLLMs on 3DSRBench are primarily sourced from its paper [25]. We then post-process the results of 3DSRBench using the official script [25]. Additionally, the performances of MLLMs on CV-Bench and BLINK are primarily derived from [30]. The average score on CV-Bench is recalculated following [37]. The results are shown in the Table. 1.

On the 3DSRBench, both knowledge injection and generalization enhancement lead to performance gains in the model. Our ultimate model showcases a harmonious balance of the various improvements, leading to optimize overall performance. Despite improvements, Actial remains slightly behind the previous state-of-the-art models on this benchmark, limited by the baseline method's performance.

While on CV-Bench, Actial achieves a substantial performance gain over the baseline and outperforms existing proprietary models. The improvements highlight the effectiveness of our proposed approach to activate the spatial reasoning capabilities of MLLMs, leading to more accurate and robust performance on visual-spatial tasks. This outcome underscores the importance of structured knowledge injection and targeted training strategies in advancing the visual reasoning abilities of large-scale models.

We mainly report the spatial tasks in BLINK following [30, 49]. The results of the multi-view component (which is similar to the subtask of our Viewpoint-100K) demonstrate that mastering foundational viewpoints is relatively straightforward for MLLMs. It provides evidence of the capability of MLLMs to perceive and reason about spatial information, highlighting their potential for advanced visual-spatial tasks. However, prior to specific activation and fine-tuning, existing

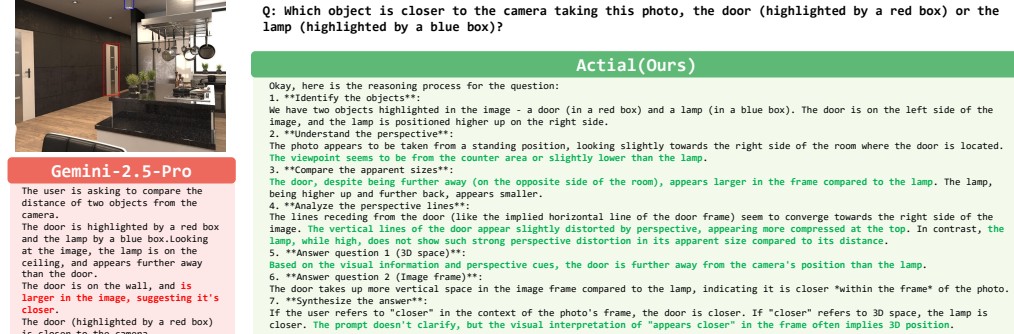

Q: Which object is closer to the camera taking this photo, the door (highlighted by a red box) or the lamp (highlighted by a blue box)?

**Actial(Ours)**

Okay, here is the reasoning process for the question:
1. **Identify the objects**:
We have two objects highlighted in the image - a door (in a red box) and a lamp (in a blue box). The door is on the left side of the image, and the lamp is positioned higher up on the right side.
2. **Understand the perspective**:
The photo appears to be taken from a standing position, looking slightly towards the right side of the room where the door is located. The viewpoint seems to be from the counter area or slightly lower than the lamp.
3. **Compare the apparent sizes**:
The door, despite being further away (on the opposite side of the room), appears larger in the frame compared to the lamp. The lamp, being higher up and further back, appears smaller.
4. **Analyze the perspective lines**:
The lines receding from the door (like the implied horizontal line of the door frame) seem to converge towards the right side of the image. The vertical lines of the door appear slightly distorted by perspective, appearing more compressed at the top. In contrast, the lamp, while high, does not show such strong perspective distortion in its apparent size compared to its distance.
5. **Answer question 1 (3D space)**:
Based on the visual information and perspective cues, the door is further away from the camera's position than the lamp.
6. **Answer question 2 (Image frame)**:
The door takes up more vertical space in the image frame compared to the lamp, indicating it is closer *within the frame* of the photo.
7. **Synthesize the answer**:
If the user refers to "closer" in the context of the photo's frame, the door is closer. If "closer" refers to 3D space, the lamp is closer. The prompt doesn't clarify, but the visual interpretation of "appears closer" in the frame often implies 3D position.

**Gemini-2.5-Pro**

The user is asking to compare the distance of two objects from the camera.
The door is highlighted by a red box and the lamp by a blue box. Looking at the image, the lamp is on the ceiling, and appears further away than the door.
The door is on the wall, and **is larger in the image, suggesting it's closer.**
The door (highlighted by a red box) is closer to the camera.

Figure 5: **The reasoning process.** Actial uses the correct spatial thinking approach.

large models perform at a level comparable to random guessing. This indicates the importance of developing foundational spatial skills for spatial reasoning.

We aim to demonstrate three key points through our experiments and address the questions raised at the beginning of this paper. First, current MLLMs have not yet fully mastered certain foundational spatial skills, such as viewpoint understanding. The random performance on BLINK's multi-view task can evident this. Second, despite being trained on large-scale 2D data, these models possess significant potential for learning 3D spatial perception. This is supported by our strong performance on the viewpoint task, achieving a score of 99.2. Third, explicitly training MLLMs on basic spatial skills can effectively enhance their spatial reasoning capabilities, leading to improved performance across a variety of tasks. Our improvement compared to baseline on multiple tasks is consistent with this point. Collectively, these findings highlight the critical importance of developing foundational spatial abilities in MLLMs as a necessary step toward enabling them to tackle more complex and nuanced visual reasoning tasks.

## 5.3 Ablation Studies

**Knowledge Injection (SFT phase).** The experimental results across most tasks indicate that knowledge injection effectively improves model performance, highlighting its beneficial impact on the learning process. Since SFT is task-specific and uses limited data diversity, it typically harms performance on tasks outside the fine-tuning distribution. This explains the occasional instances of marginally lower performance (e.g., the Orient. in 3DSRBench and the RelDep in Blink) compared to the ablation model. However, we were pleasantly surprised to find that fine-tuning on viewpoint tasks led to performance improvements on out-of-domain tasks, such as Height, Depth, and Relation. This highlights the importance of viewpoint learning for enhancing the model's overall spatial ability.

**Generalization Enhancement (GRPO phase).** The evaluation across various benchmarks demonstrates that although our knowledge injection approach enhances model performance on specific tasks, the homogeneity of these tasks and the training methods can result in performance degradation on others. The subsequent generalization enhancement phase not only retains the improvements achieved through knowledge injection but also effectively addresses the observed performance declines. Moreover, this phase facilitates additional performance gains on tasks where the model initially demonstrated strong capabilities, thereby achieving even greater improvement. However, we also observed that generalization enhancement can lead to a decrease in metrics for some tasks compared to the knowledge injection stage. For instance, on 3DSRBench, knowledge injection improved the model's understanding of height and location, but after generalization enhancement, performance in these areas declined while improving in two other tasks. This suggests that relying solely on result-based rewards can still affect the previously injected foundational knowledge to some extent.

# 6 Conclusion, Limitation and Impact

This study aims to activate the spatial reasoning ability within Multimodal Large Language Models. Motivated by the need to bridge the gap between 2D visual understanding and robust 3D spatial reasoning, we introduce Viewpoint Learning, a task designed to evaluate and improve MLLMs' spatial reasoning abilities. We employ a two-stage fine-tuning strategy: first, SFT with hybrid cold-start initialization on Viewpoint-100K injects foundational knowledge, followed by Reinforcement Learning using the GRPO algorithm to enhance generalization. Our results show that this approach significantly improves the model's performance in both in-domain and out-of-domain reasoning tasks, demonstrating a meaningful activation of its spatial reasoning ability. Although current MLLMs lack an explicit understanding of 3D geometry, our findings indicate that targeted training strategies can effectively unlock their potential for spatial reasoning. However, the scenarios and tasks included in our dataset are relatively constrained, with all data being object-centric, which simplifies the problems compared to more varied and complex settings. The tasks in our dataset primarily address the basic aspects of Viewpoint Learning, when compared with more challenging tasks such as camera pose estimation. Cultivating these foundational spatial skills is crucial for advancing MLLMs towards more complex visual tasks. This work provides a practical pathway for improving 3D perception in MLLMs, with direct applications in robotics, autonomous navigation, and 3D scene understanding.

## Acknowledgments

This work is partly supported by the National Natural Science Foundation of China (62032011) and the Natural Science Foundation of Jiangsu Province (BK20211147).

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

# Appendix

We display the additional experiment results in Section. A and analyze the training process in Section. B. We further present dataset examples and the utilized prompts in Section. C and Section. D, respectively.

## A  Additional Experiments

### A.1  Results on Viewpoint-100K

The test set of Viewpoint-100K consists of 1,000 examples. To assess human performance, we randomly sampled 100 instances from the test set and conducted a human evaluation with three annotators, resulting in an average accuracy of 97.67%. This high level of performance can be attributed to our deliberate design choices during data construction; specifically, to ensure a reasonable level of task difficulty, we restricted the angular difference between image pairs to a range of ±20 to ±100 degrees, which maintains a challenging yet discriminable task for both humans and models. In comparison, the Qwen base model achieved an accuracy of only 12.9%, indicating limited capability in handling the viewpoint estimation task without further training. After SFT, the model's accuracy improved significantly to 92.2%, demonstrating the effectiveness of training on labeled exemplars. However, when GRPO was applied following SFT, the performance decreased to 81.4%, suggesting that the RL objective may not align well with the downstream task or that the reward signal requires further refinement.

### A.2  Results on MMSI-Bench

To further illustrate the effectiveness and generalization capability of our approach on more complex tasks, we conduct additional experiments on the more comprehensive and challenging MMSI-Bench [46]. Due to our long CoT template length, we set the max tokens to 32K. As shown in Table. A1, Actial achieves comprehensive improvements in many subtasks when compared to the baseline model. Moreover, our model, with only 7B parameters, achieves performance comparable to that of larger models and GPT-4o, and even outperforms them on certain tasks. The significant difference on the MSR task is due to Actial generating excessively long reasoning chains when handling multi-step inference, causing the output to be truncated before the correct answer is reached. This prevents the model from producing complete and accurate responses. Addressing this issue by optimizing reasoning efficiency and managing output length will be an important direction for future work.

### A.3  Additional Ablation Study

We conducted this additional ablation experiment by mixing Viewpoint-100K, pseudo CoTs, and SAT. We only use SFT (without GRPO) to fine-tune Qwen2.5-VL-7B-Instruct using the same training parameters as in the previous experiments. We trained for a total of 2,000 steps (approximately 1.5 epochs). The ablation results are presented in the bottom section of Table. A1. The significant gap demonstrates the effectiveness of our two-stage training framework and highlight the significant performance gains achieved through GRPO in OOD tasks.

Table A1: **Evaluation on MMSI-Bench.** We color the best and the second-best results. ↑ indicates improvement over the baseline.

| MMSI-Bench | Cam.-Cam. | Obj.-Obj. | Reg.-Reg. | Cam.-Obj. | Obj.-Reg. | Cam.-Reg. | Means. | Appr. | Motion-Cam. | Motion-Obj. | MSR | Avg. |
|---|---|---|---|---|---|---|---|---|---|---|---|---|
| GPT-4o | 34.4 | 24.5 | 23.5 | 19.8 | 37.6 | 27.7 | 32.8 | 31.8 | 35.1 | 36.8 | 30.8 | 30.3 |
| Qwen2.5-VL-72B | 25.8 | 34.0 | 34.6 | 23.3 | 34.1 | 36.1 | 45.3 | 27.3 | 27.0 | 30.3 | 27.3 | 30.7 |
| InternVL2.5-78B | 23.7 | 22.3 | 39.5 | 29.1 | 31.8 | 42.2 | 35.9 | 19.7 | 17.6 | 26.3 | 27.3 | 28.5 |
| Qwen2.5-VL-7B-Instruct | 23.7 | 24.5 | 19.8 | 25.6 | 32.9 | 33.7 | 42.2 | 24.2 | 18.9 | 30.3 | 23.2 | 26.5 |
| Actial-7B(Ours) | 29.0 ↑ | 31.9 ↑ | 28.4↑ | 41.9 ↑ | 28.2 | 33.7 | 31.3 | 22.7 | 27.0 ↑ | 32.9 ↑ | 20.7 | 28.9↑ |
| Ablation Model | 19.4 | 29.8↑ | 27.2↑ | 31.4 ↑ | 30.6 | 34.9↑ | 29.7 | 19.7 | 25.7↑ | 25.0 | 26.3↑ | 27.2↑ |

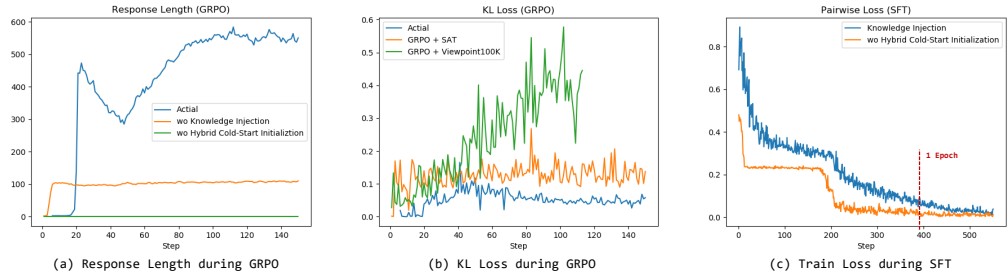

Figure A1: **Metrics changes during the training process.**

## B  Training Analysis

Figure A1(a) presents the evolution of the model's response length throughout the GRPO phase. Our observations align with those reported in [13], where Actial displays an initial reduction followed by a subsequent adjustment in response length. When contrasted with direct fine-tuning of the baseline model, Actial demonstrates an extended reasoning length. The green line (without hybrid cold-start initialization) reflects no significant increase in response length, attributable mainly to the Supervised Fine-Tuning (SFT) stage. This stagnation can be explained by the composition of our Viewpoint-100K dataset, which comprises exclusively multiple-choice questions devoid of reasoning templates. Consequently, the model struggles to accurately achieve format rewards during the GRPO process, and we introduce the hybrid cold-start initialization to improve such issue.

Figure A1(b) depicts the evolution in KL Divergence across different variants. The green line, which represents the direct application of Viewpoint-100K for GRPO, exhibits a growing offset relative to the initial strategy, suggesting that the baseline model's original spatial reasoning capabilities are inadequate for handling viewpoint-specific tasks. Conversely, utilizing the SAT dataset leads to substantially lower KL divergence, underscoring the unique spatial reasoning demands posed by our Viewpoint-100K dataset. Following knowledge injection, the KL divergence becomes notably smoother, indicating the efficacy of integrating foundational viewpoint knowledge.

Figure A1(c) illustrates the pairwise loss observed during the Supervised Fine-Tuning (SFT) training phase. Notably, when using our Viewpoint-100K dataset, there is a sudden and significant improvement in performance (a trend also reflected in the validation curve). This rapid decrease excludes the possibility of the model simply memorizing the answers since it appears within one single epoch. Additionally, when using a hybrid cold-start initialization, which requires the model to learn reasoning templates, the loss curve becomes more smoother. However, the sudden insight remains clearly evident, reflecting its relevance to our viewpoint-based questions. We believe that this phenomenon mainly comes from two reasons. First, our dataset consists of relatively simple multiple-choice questions with only three options (one of which is a distractor), making it easy for the model to select the correct answer even without proper reasoning. However, such correctness achieved through flawed reasoning is insufficient for the model to truly understand and solve viewpoint-related problems, leading to a oscillation phase of loss fluctuation. Second, as mentioned in the main text, existing MLLMs tend to rely on incorrect 2D cues when solving 3D tasks. In contrast, viewpoint problems require the model to learn how to properly utilize 3D spatial cues, resulting in a period where the loss remains stagnant. Nevertheless, MLLMs do possess latent 3D perception capabilities. Once the model learns to shift its perspective appropriately (seems to be like the activation), it can rapidly generalize this understanding to similar tasks, leading to a sudden drop in loss.

## C  Dataset Examples

We show the examples of Viewpoint-100K in Figure. A3. We provide three types of questions, including the horizontal translation and rotation from the camera's perspective and the horizontal translation from the object's perspective. We also provide the accurate rotation angles in our dataset, calculated from the camera parameters.

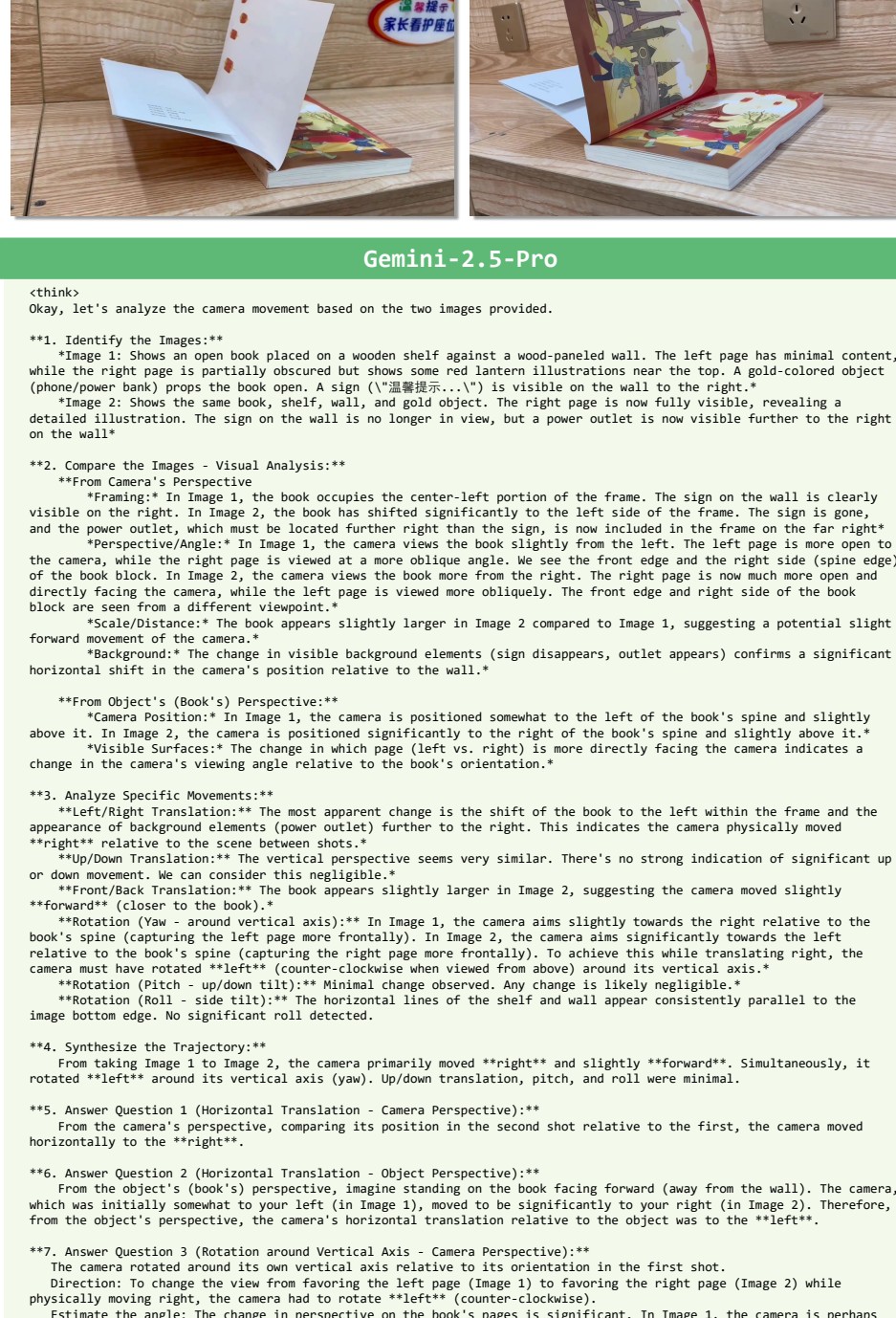

```
<think>
Okay, let's analyze the camera movement based on the two images provided.

**1. Identify the Images:**
    *Image 1: Shows an open book placed on a wooden shelf against a wood-paneled wall. The left page has minimal content,
while the right page is partially obscured but shows some red lantern illustrations near the top. A gold-colored object
(phone/power bank) props the book open. A sign (\"温馨提示...\") is visible on the wall to the right.*
    *Image 2: Shows the same book, shelf, wall, and gold object. The right page is now fully visible, revealing a
detailed illustration. The sign on the wall is no longer in view, but a power outlet is now visible further to the right
on the wall*

**2. Compare the Images - Visual Analysis:**
    **From Camera's Perspective
        *Framing:* In Image 1, the book occupies the center-left portion of the frame. The sign on the wall is clearly
visible on the right. In Image 2, the book has shifted significantly to the left side of the frame. The sign is gone,
and the power outlet, which must be located further right than the sign, is now included in the frame on the far right*
        *Perspective/Angle:* In Image 1, the camera views the book slightly from the left. The left page is more open to
the camera, while the right page is viewed at a more oblique angle. We see the front edge and the right side (spine edge)
of the book block. In Image 2, the camera views the book more from the right. The right page is now much more open and
directly facing the camera, while the left page is viewed more obliquely. The front edge and right side of the book
block are seen from a different viewpoint.*
        *Scale/Distance:* The book appears slightly larger in Image 2 compared to Image 1, suggesting a potential slight
forward movement of the camera.*
        *Background:* The change in visible background elements (sign disappears, outlet appears) confirms a significant
horizontal shift in the camera's position relative to the wall.*

    **From Object's (Book's) Perspective:**
        *Camera Position:* In Image 1, the camera is positioned somewhat to the left of the book's spine and slightly
above it. In Image 2, the camera is positioned significantly to the right of the book's spine and slightly above it.*
        *Visible Surfaces:* The change in which page (left vs. right) is more directly facing the camera indicates a
change in the camera's viewing angle relative to the book's orientation.*

**3. Analyze Specific Movements:**
    **Left/Right Translation:** The most apparent change is the shift of the book to the left within the frame and the
appearance of background elements (power outlet) further to the right. This indicates the camera physically moved
**right** relative to the scene between shots.*
    **Up/Down Translation:** The vertical perspective seems very similar. There's no strong indication of significant up
or down movement. We can consider this negligible.*
    **Front/Back Translation:** The book appears slightly larger in Image 2, suggesting the camera moved slightly
**forward** (closer to the book).*
    **Rotation (Yaw - around vertical axis):** In Image 1, the camera aims slightly towards the right relative to the
book's spine (capturing the left page more frontally). In Image 2, the camera aims significantly towards the left
relative to the book's spine (capturing the right page more frontally). To achieve this while translating right, the
camera must have rotated **left** (counter-clockwise when viewed from above) around its vertical axis.*
    **Rotation (Pitch - up/down tilt):** Minimal change observed. Any change is likely negligible.*
    **Rotation (Roll - side tilt):** The horizontal lines of the shelf and wall appear consistently parallel to the
image bottom edge. No significant roll detected.

**4. Synthesize the Trajectory:**
    From taking Image 1 to Image 2, the camera primarily moved **right** and slightly **forward**. Simultaneously, it
rotated **left** around its vertical axis (yaw). Up/down translation, pitch, and roll were minimal.

**5. Answer Question 1 (Horizontal Translation - Camera Perspective):**
    From the camera's perspective, comparing its position in the second shot relative to the first, the camera moved
horizontally to the **right**.

**6. Answer Question 2 (Horizontal Translation - Object Perspective):**
    From the object's (book's) perspective, imagine standing on the book facing forward (away from the wall). The camera,
which was initially somewhat to your left (in Image 1), moved to be significantly to your right (in Image 2). Therefore,
from the object's perspective, the camera's horizontal translation relative to the object was to the **left**.

**7. Answer Question 3 (Rotation around Vertical Axis - Camera Perspective):**
   The camera rotated around its own vertical axis relative to its orientation in the first shot.
   Direction: To change the view from favoring the left page (Image 1) to favoring the right page (Image 2) while
physically moving right, the camera had to rotate **left** (counter-clockwise).
   Estimate the angle: The change in perspective on the book's pages is significant. In Image 1, the camera is perhaps
looking 20-30 degrees to the right of the book's spine normal. In Image 2, it's looking perhaps 20-30 degrees to the
left of the spine normal. The total change in angle relative to the book is substantial. Estimating this change based on
how much more \"open\" the right page becomes suggests a rotation of approximately **54 degrees** to the left.
</think>
<answer>A</answer>
```

Figure A2: **An example of our generated pseudo CoT.**

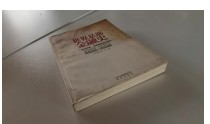
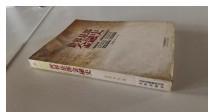

1.`<image><image>`The object captured in these two photos did not move. From the camera's perspective, did the position of the camera that captured the second photo move relative to the first photo's camera position?

A: Move to the left  B: Move to the right  C: No movement

Select the correct option, please output only the option letter!

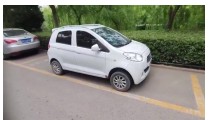
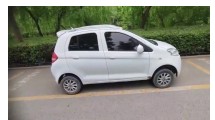

2.`<image><image>`The object captured in these two photos did not move. From the object's perspective, did the position of the camera that captured the second photo move relative to the first photo's camera position?

A: Move to the right  B: No movement  C: Move to the left

Select the correct option, please output only the option letter!

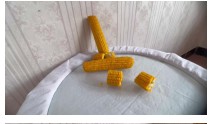
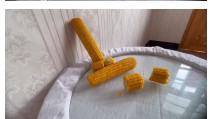

3.`<image><image>`The object captured in these two photos did not move. From the camera's perspective, did the orientation of the camera that captured the second photo rotate relative to the first photo's camera orientation?

A: No rotation  B: Rotate to the right  C: Rotate to the left

Select the correct option, please output only the option letter!

Figure A3: **The QA examples of Viewpoint-100K.** We provide three types of questions.

## D  Prompts

Our system prompt is copied from VLAA [6], shown as follow:

> **System Prompt:**
>
> You are Actial, a helpful assistant with excellent reasoning ability.
> A user asks you a question, and you should try to solve it.
> You should first think about the reasoning process in the mind and then provides the user with the answer.
> The reasoning process and answer are enclosed within <think> </think> and <answer> </answer> tags, respectively, i.e., <think> reasoning process here </think> <answer> answer here </answer>"

We show an example of our generated pseudo CoT in Figure. A2. The human-assisted CoT template is as follow:

> **Human-Assisted CoT Template:**
>
> Based on the content presented in the two pictures, analyze the movement trajectory of the camera from shooting the first picture to shooting the second one, Describe the movement trajectory specifically, including front, back, left, right, up and down, as well as rotation . and then answer the following two questions based on the camera trajectory.
>
> 1. From the camera's perspective, determine the horizontal translation of the second camera relative to the first camera.
> 2. From the object's perspective, determine the horizontal translation of the second camera

relative to the first camera.
3. From the camera's perspective, determine the rotation of the second camera around its own verticalaxis relative to the first camera's orientation:
- Estimate the rotationangle in degrees.

Thinking:
I need to get the relevant content to answer the user's question.
Here's a thinking process to analyze the camera movement between the two images:

Understand the Goal: The user wants to know how the camera moved between taking the first picture (1.jpg) and the second picture (2.jpg). The object did not move in any way from the first picture to the second. The description needs to cover translation (front/back, left/right, up/down) and rotation. Then, I need to answer two specific questions: horizontal translation (left/right/none) and rotation angle around the camera's vertical axis.

1.Identify the Images:
Image 1: 1.jpg (Let's call this Pic 1)
Image 2: 2.jpg (Let's call this Pic 2)
The object in both pictures is a red, quilted handbag with a gold chain strap, placed on a blue-and-white striped surface with yellow stars and pink crescents.

2.Compare the Images - Visual Analysis:

From camera's perspective:

Frames:
In Pic 1, we see more of the surface above the bag. From a camera centered perspective, we can see the left plane of the bag, but we cannot see the right plane of the bag.
In Pic 2, from a camera-centered perspective, the front of the bag is visible, with the golden chain strap and logo pointing toward the bottom of the image. The left side is not visible.

Perspective/Angle:
In Pic 1, from the camera's perspective, shows more of the front and the left side of the bag. The chain strap is clearly visible and faces towards the bottom left of the photo.
In Pic 2, from the camera's perspective, shows more of the front side of the bag. The clasp is still visible but from a slightly different angle, it faces towards the bottom of the photo. The top surface of the bag seems a bit more visible in Pic 2, suggesting a slightly higher viewpoint or a slight downward tilt.

From object's perspective:

Frames:
In Pic 1, define the side of the bag featuring the gold chain strap and logo as its front. From the bag's perspective, the camera is positioned slightly to the left and above the bag, with a viewpoint angled downward and toward the left plane of the bag.
In Pic 2, from the bag's perspective, the camera moves to the right. It is now positioned slightly in front of and above the bag, with a slight downward angle.

Perspective/Angle:
In Pic 1, from the bag's perspective, both the front and left planes are visible to the camera.
In Pic 2, from the bag's perspective, the left plane is no longer visible, indicating that the camera has rotated to face more directly in front of the bag.

Scale/Distance: The bag appears roughly the same size in both images, suggesting minimal movement towards or away from the object (front/back translation).

Background: The wall/surface behind the striped fabric looks similar in both, although the angle relative to the bag changes.

3.Analyze Specific Movements:
Left/Right Translation: If the camera undergoes only horizontal translation, we can compare the position of the bag relative to the image frame. However, in this case, the camera trajectory involves rotation as well. In Pic 1, both the front and left planes of the bag are visible. In Pic 2, the left plane is no longer visible, while the front plane remains in view. The change in visible surfaces indicates that the camera must have moved to the left relative to the bag.

Up/Down Translation: In Pic 1, there's more space above the bag. In Pic 2, there's less space above the bag, and the camera orientation seems slightly higher relative to the bag's top surface. This suggests the camera moved slightly up.

Front/Back Translation: The scale of the bag seems very similar. There's no strong indication of significant movement towards or away from the bag. Let's assume minimal or no front/back translation.

Rotation (Yaw - around vertical axis): In Pic 1, the front plane of the bag faces toward the bottom-left of the photo. In Pic 2, the front plane faces toward the bottom of the photo. Given the leftward horizontal translation of the camera, to achieve this change in perspective while keeping the bag relatively stationary, the camera must have rotated rightward (clockwise when viewed from above) around its vertical axis.

Rotation (Pitch - up/down tilt): The top surface seems slightly more visible in Pic 2. This could be due to the upward translation or a slight downward tilt (pitch down). Given the upward translation seems likely, a slight downward tilt might also be present, but the upward movement is more apparent. Let's focus on the dominant movements.

Rotation (Roll - side tilt): The horizontal lines of the striped fabric and the wall seem roughly parallel to the image edges in both pictures. There's no significant roll.

4.Synthesize the Trajectory:
From Pic 1 to Pic 2, the camera moved primarily left and slightly up. It also rotated rightward around its vertical axis (yaw). There was likely minimal front/back movement and minimal roll or pitch changes.

5.Answer Question 1 (Horizontal Translation in camera's perspective):
The camera moved horizontally relative to the bag.
Did it move left or right? It moved left.
The question asks for the horizontal translation of the second camera relative to the first camera. The answer is left.

6.Answer Question 2 (Horizontal Translation in object's perspective):
Consider the object's perspective, the camera now moved to the right. Did it move left or right? It moved right.
The question asks for the horizontal translation of camera in object's perspective.
The key point of the problem is to transform the reference frame onto the object. The answer is left.

7.Answer Question 3 (Rotation around Vertical Axis in camera's perspective):
The camera rotated around its vertical axis relative to its orientation in the second shot.

Did it rotate left or right? It rotated rightward.
Estimate the angle. Looking at the changes in visible parts of objects in the two photos, a significant rotation occurred. Comparing the visible faces, the first photo includs the front

and left part of the bag in the camera's perspective. In Pic 2, we see the front almost squarely, maybe slightly revealing the right edge. It looks like roughly a 80 to 90-degree change in viewing angle relative to the bag.

Did it rotate to the left or right? Since the original left part of the bag in the first photo can not be seen in the second photo, it represents the camera rotates to the right, resulting in more visible part of the left side of the front of the bag.
Estimate: Let's go with approximately 87 degrees. (This is an estimation based on the visual change). The answer is the camera rotates 87 degrees to the right.

7.Format the Output: Combine the trajectory description and the answers to the specific questions clearly. Ensure the answers use the specified format.

The prompt used to generate the pseudo CoTs is as follow:

**Pseudo CoT Generation:**

*<image><image>*

I'll give you an example, please follow the example to analyze the camera movement between the two images.

[*Human-Assisted CoT Template*]

Please analyze the given two images using the same approach as I provided above. Including the Image Identify, Visual Analysis, Analyze Specific Movements, Synthesize the Trajectory, Answer Question.

Do not use the relative position changes of objects in the image, consider the changes in three-dimensional space.

The correct answer is: move to the *gt[0]* in cameraś perspective; move to the *gt[1]* in objectś perspective; rotate to the *gt[2]* with about *gt[3]* degrees. Please provide a sufficiently detailed analysis..

