# OpenReview forum: "Actial: Activate Spatial Reasoning Ability of Multimodal Large Language Models"
_NeurIPS.cc/2025/Conference — NeurIPS 2025 poster_

### Official Review · Reviewer_hxxQ · 2025-06-23

**Clarity:** 3
**Significance:** 3
**Originality:** 3
**Rating:** 5
**Confidence:** 3

**Summary:**

This paper introduces a new task, namely viewpoint learning, to improve the spatial reasoning capabilities of 2D MLLMs. Specifically, this work injects spatial knowledge by finetuning the MLLM on the viewpoint learning tasks using a visual QA dataset curated for that specific task. Further, to improve the generalization capabilities, GRPO is used to train the fonetuned LLM on a broader set of questions using synthetic data.

**Questions:**

See weaknesses

**Ethical Concerns:**

["NO or VERY MINOR ethics concerns only"]

**Final Justification:**

The authors have addressed my concerns and I lean towards acceptance of this paper. So I will maintain my rating

**Limitations:**

yes

**Quality:**

3

**Strengths And Weaknesses:**

Strengths:-

- The paper is very well-written with a clear motivation.
- This is the first work to identify and demonstrate that finetuning MLLMs on viewpoint learning significantly boosts their spatial reasoning capabilities.
- The idea of using GRPO for enhancing generalization after finetuning is interesting.
- Extensive experiments have been conducted to demonstrate SOTA performances on multiple benchmark datasets.

Weaknesses:-

- Recent works such as MLLM-for3D incorporates a similar idea of having cross-view spatial consistency using multi-view segmentation maps to enhance reasoning capabilities of 2D MLLMs. Comparison with those methods could be more helpful in assessing the efficacy of your method.
- The method claims to enhance out-of-domain generalization but does not explicitly showcase this in the results section. A better expanation on this would be helpful.

Additional points:-

- Spatial-MLLM seems to be a concurrent work that follows a very similar pipeline of Supervised fine-tuning on a curated QA dataset followed by GRPO. However, their QA pairs cover different spatial reasoning tasks such as object counting, object size, relative distance etc. Using those tasks along viewpoint consistency could be more beneficial for the initial finetuning of MLLMs.

---

> ### Author Rebuttal · Authors · 2025-07-30
>
> ## **Q1: Comparison with MLLM-for3D and Spatial-MLLM.**
>
> **Response:** Thank you for drawing our attention to these excellent concurrent works. We will carefully compare and discuss them in the revised manuscript.
>
> As mentioned in the `Related Work` section, both `MLLM-for3D` and `Spatial-MLLM` belong to a line of research that enhances the integration of MLLMs with 3D environments by introducing `additional 3D inputs and features`, aiming for more reliable 3D understanding and reasoning.
>
> - `MLLM-for3D` further advances this idea by `lifting` 2D segmentation features from multiple views into 3D space through a spatial consistency-guided fusion mechanism, enabling direct 3D spatial reasoning. The key difference from our work is that MLLM-for3D treats 3D consistency as an `external prior` to align 2D reasoning results across views in 3D space. In contrast, **our final goal is to investigate whether an MLLM can internalize and actively use the concept of 3D consistency within its own reasoning process, and to evaluate how such internalization affects its spatial understanding.**
>
> - Regarding `Spatial-MLLM`, there are two main distinctions:
>
>     - First, Spatial-MLLM aims to improve spatial understanding through high-quality, diverse multi-task data and `additional 3D features` introduced via a `VGGT backbone`. In contrast, our work focuses on assessing the impact of a single, carefully designed task (Viewpoint Learning) on an MLLM's 3D spatial reasoning. **Our goal is to highlight the importance of task design and data selection in activating spatial reasoning abilities, offering insights for future work.**
>
>     - Second, in Spatial-MLLM, both the SFT and GRPO stages are designed to improve performance on the Spatial-MLLM-120K benchmark. In our approach, the SFT stage is specifically used to inject viewpoint-related knowledge, while the GRPO stage aims to mitigate the potential degradation in generalization caused by task-specific fine-tuning. More importantly, it encourages the model to transfer the spatial reasoning skills (like the reasoning process shown in `Figure 5` in the manuscript) learned from the viewpoint task to OOD tasks.
>
> These differences reflect our distinct goal: **to understand and enhance how MLLMs reason about 3D space relying solely on 2D visual inputs.**
>
> ## **Q2: More explanations about the OOD generalization.**
>
> **Response:** Thank you for highlighting this part. We will carefully address and revise the manuscript accordingly.
>
> In this paper, the generalization ability showcases on three aspects.
>
> - First, in our SFT phase, we fine-tuned the MLLM exclusively on viewpoint tasks. As typical with `task-specific fine-tuning without diverse data`, this can harm performance on out-of-domain tasks, such as the drop we observe on Orient. (3DSRBench) compared to the base model. Encouragingly, however, we find that even with fine-tuning on just this single task, the model shows `performance gains` on several spatial reasoning tasks (e.g., Height, Relation, Depth). This suggests that **mastering viewpoint reasoning can generalize broadly and significantly enhance the model's overall spatial reasoning ability.**
>
> - Second, it refers to **generalize reasoning capabilities to OOD tasks.** Since our CoT templates only include the viewpoint task. While after GRPO, the model can correctly transfer spatial reasoning to a diverse range of OOD tasks.
>
> - Finally, to further illustrate the effectiveness and generalization capability of our approach on more complex tasks, we conduct additional experiments on the `more comprehensive and challenging` **MMSI-Bench**[1]. Due to our long CoT template length, we set the max_tokens to 32K.
>
>
>     | MMSI-Bench | Param | Cam.-Cam. | Obj.-Obj. | Reg.-Reg. | Cam.-Obj. | Obj.-Reg. | Cam.-Reg. | Meas. | Appr. | Motion-Cam. | Motion-Obj. | MSR | Avg. |
>     | -- | -- | -- | -- | -- | -- | -- | -- | -- | -- | -- | -- | -- | -- |
>     | GPT-4o | - | **34.4** | 24.5 | 23.5 | 19.8 | **37.6** | 27.7 | 32.8 | **31.8** | **35.1** | **36.8** | **30.8** | 30.3 |
>     | Qwen2.5-VL-72B | 72B |25.8 | **34.0** | 34.6 | 23.3 | 34.1 | 36.1 | **45.3** | 27.3 | 27.0 | 30.3 | 27.3 | **30.7** |
>     | InternVL2.5-78B | 78B | 23.7 | 22.3 | **39.5** | 29.1 | 31.8 | **42.2** | 35.9 | 19.7 | 17.6 | 26.3 | 27.3 | 28.5 |
>     | Qwen2.5-VL-7B-Instruct | 7B | 23.7 | 24.5 | 19.8 | 25.6 | 32.9 | 33.7 | 42.2 | 24.2 | 18.9 | 30.3 | 23.2 | 26.5 |
>     | **Actial-7B(Ours)** | 7B | 29.0 `(+5.3)` | 31.9 `(+7.4)` | 28.4 `(+8.6)` | **41.9** `(+16.3)` |28.2 (-4.7) | 33.7 | 31.3 (-10.9) | 22.7 (-1.5) | 27.0 `(+8.1)` | 32.9 `(+2.6)` | 20.7 (-2.5) | 28.9 `(+2.4)` |
>     | **Ablation Model** | 7B | 19.4 `(-9.6)` | 29.8 `(-2.1)` | 27.2 `(-1.2)` | 31.4 `(-10.5)` | 30.6 (+2.4) | 34.9 (+1.2) | 29.7 `(-1.6)` | 19.7 `(-3.0)` |  25.7 `(-1.3)` |  25.0 `(-7.9)` | 26.3 (+5.6) | 27.2 `(-2.6)` |
>
>     Compared to the baseline model, **Actial** achieves comprehensive improvements in many subtasks. Moreover, our model, with `only 7B parameters`, achieves performance comparable to that of larger models and GPT-4o, and even outperforms them on certain tasks.
>
>    The `Ablation Model` was trained in accordance with the suggestion provided by `Reviewer AHFe`. We conducted this additional ablation experiment by mixing Viewpoint-100K, pseudo CoTs, and SAT. We use SFT `(without GRPO)` to fine-tune Qwen2.5-VL-7B-Instruct using the same training parameters as in the previous experiments. We trained for a total of 2,000 steps (approximately 1.5 epochs). The significant gap demonstrates the effectiveness of our two-stage training framework and highlight the significant performance gains achieved through GRPO in OOD tasks.
>
>     And the difference on the `MSR task` is due to Actial generating excessively `long reasoning chains` when handling multi-step inference, causing the output to `be truncated` before the correct answer is reached. This prevents the model from producing complete and accurate responses. Addressing this issue by optimizing reasoning efficiency and managing output length will be an important direction for future work.
>
> [1] Yang et al. 2025. MMSI-Bench: A Benchmark for Multi-Image Spatial Intelligence

---

> > ### Comment · Reviewer_hxxQ · 2025-08-04
> >
> > The authors have adequately addressed my concerns. One minor suggestion for the final version would be to include a quantitative comparison with Spatial-MLLM and MLLM-for-3D. This could further strengthen the work by demonstrating the effectiveness of the proposed method using only 2D visual inputs.

---

> > > ### Author Response · Authors · 2025-08-05
> > >
> > > We sincerely appreciate your positive feedback and valuable comments. Thank you for your time and consideration. In the revised version of the paper, we will include additional discussions and comparisons.

---

### Official Review · Reviewer_jwwJ · 2025-06-30

**Clarity:** 3
**Significance:** 2
**Originality:** 2
**Rating:** 4
**Confidence:** 3

**Summary:**

This paper introduces Actial-7B, a MLLM that is finetuned using SFT and GRPO with Viewpoint-100K dataset. The resulting model shows better performance on spatial-reasoning benchmarks, which indicates the usefulness of the proposed dataset and training pipeline. The paper is built upon the hypothesis that understanding viewpoints is an important aspect for MLLMs to understand the 3D spatial relations, which is proved by finetuning the model using Viewpoint-100K dataset, a dataset that solely consists of viewpoint understanding.

**Questions:**

1. How can viewpoint learning boost the understanding of more complex problems like orientation, as suggested by the marginal improvements on 3DSRBench?

2. Why does the author not consider the viewpoint changes for up and down, the resulting distribution of the object pose under this setting would be different from left-and-right. Is the image pairs in the dataset diverse enough to cover different angles of the objects?

3. The dataset proposed consider a fixed object and a moving camera, can the model better understand the setting where camera is fixed and object is moving?

**Ethical Concerns:**

["NO or VERY MINOR ethics concerns only"]

**Final Justification:**

The author has addressed all my concerns.

**Limitations:**

Yes

**Quality:**

2

**Strengths And Weaknesses:**

Strengths:

1. The paper proposed a useful dataset Viewpoint-100K, which can be widely used by the community to enforce better spatial relation understanding of MLLMs.

2. The finetuning recipe is clear enough to be followed.

3. The resulting model from the above two bullet points show decent performance improvements on CV-Bench and BLINK.

Weaknesses:

1. The training recipe and the dataset is only used to finetune one model, i.e.  Qwen2.5-VL-7B-Instruct, lacking a comprehensive evaluation on how the proposed method can be applied to different MLLMs.

2. The model's improvements on 3DSRBench is marginal, which indicates that for complex tasks in 3DSRBench, the model needs abilities more than just viewpoint understanding, which contradicts the authors' hypothesis that viewpoint understanding is important for 3D spatial understanding.

---

> ### Author Rebuttal · Authors · 2025-07-30
>
> ## **Q1: Viewpoint Learning for complex tasks.**
> Thanks for your valuable comments on our work. The marginal overall improvement in 3DSRBench is primarily attributed to the Orient. task, **while all other tasks demonstrate significant performance gains.**
>
>
> - **Additional experiments on complex tasks**.
>
>     To further illustrate the effectiveness and generalization capability of our approach on more complex tasks, we conduct additional experiments on the `more comprehensive and challenging` **MMSI-Bench**[1].
>
>     | MMSI-Bench | Param | Cam.-Cam. | Obj.-Obj. | Reg.-Reg. | Cam.-Obj. | Obj.-Reg. | Cam.-Reg. | Meas. | Appr. | Motion-Cam. | Motion-Obj. | MSR | Avg. |
>     | -- | -- | -- | -- | -- | -- | -- | -- | -- | -- | -- | -- | -- | -- |
>     | GPT-4o | - | **34.4** | 24.5 | 23.5 | 19.8 | **37.6** | 27.7 | 32.8 | **31.8** | **35.1** | **36.8** | **30.8** | 30.3 |
>     | Qwen2.5-VL-72B | 72B |25.8 | **34.0** | 34.6 | 23.3 | 34.1 | 36.1 | **45.3** | 27.3 | 27.0 | 30.3 | 27.3 | **30.7** |
>     | InternVL2.5-78B | 78B | 23.7 | 22.3 | **39.5** | 29.1 | 31.8 | **42.2** | 35.9 | 19.7 | 17.6 | 26.3 | 27.3 | 28.5 |
>     | Qwen2.5-VL-7B-Instruct | 7B | 23.7 | 24.5 | 19.8 | 25.6 | 32.9 | 33.7 | 42.2 | 24.2 | 18.9 | 30.3 | 23.2 | 26.5 |
>     | **Actial-7B(Ours)** | 7B | 29.0 `(+5.3)` | 31.9 `(+7.4)` | 28.4 `(+8.6)` | **41.9** `(+16.3)` |28.2 (-4.7) | 33.7 | 31.3 (-10.9) | 22.7 (-1.5) | 27.0 `(+8.1)` | 32.9 `(+2.6)` | 20.7 (-2.5) | 28.9 `(+2.4)` |
>
>     Compared to the baseline model, **Actial** achieves comprehensive improvements in many subtasks. Moreover, our model, with `only 7B parameters`, achieves performance comparable to that of larger models and GPT-4o, and even outperforms them on certain tasks.
>
> [1] Yang et al. 2025. MMSI-Bench: A Benchmark for Multi-Image Spatial Intelligence
>
> - The explanation of the drop on `Orient.` task (3DSRBench).
>     - We have carefully checked the examples of Orient. task in 3DSRBench. As shown in their paper, the question forms like `'Consider the real-world 3D locations and orientations of the objects. Is the stop sign on the left or right side of the man on the bicycle?'`. And this remains a crucial problem that **the question does not specify the frames of reference (which we carefully descripted in Viewpoint-100K dataset)**.  In general, different people may adopt different default frames of reference depending on the context, so without explicitly specifying the reference frame, these questions are difficult to answer "correctly". As a result, the question formulations in 3DSRBench significant challenges for our model.
>     - This is also why we introduce a `second-stage GRPO`, aiming to enhance the model's generalization capability through reinforcement learning on a larger and more diverse dataset. Indeed, after RL, we observe significant performance `improvements across the majority of tasks` compared to the SFT phase ("w/o G.E."), while the results on BLINK indicate that the viewpoint knowledge injected during SFT is `well preserved`. **These findings are consistent with our core argument.**
>     - However, the suboptimal performance on certain tasks may be attributed to several factors:
>         - First, SAT is a synthetic dataset, and there is a significant `domain gap` between its rendered images and real-world visuals, along with relatively low data quality.
>         - Second, there remains room for improvement in our RL training process. Due to the `strong influence of the SFT phase`, the model's performance on some tasks has not yet reached its full potential.
>         - We believe this can be addressed by further incorporating `higher-quality real-world data` to enhance the second-stage RL and improve generalization.
>
>
>
>
> ## **Q2.1: Up-down viewpoint questions.**
>
> **Response:** Thank you for your insightful question. This is a very important question in 3D domain. **We treat changes in up-down viewpoint as a dual problem to left-right changes.** In 3D space, the three coordinate axes are `orthogonal` and `independent`. Therefore, if a model can distinguish left-right transformations, we expect it to have the `potential` to distinguish up-down changes as well. In fact, rotating an up-down image pair by 90 degrees makes it equivalent to a left-right pair (this equivalence arises from the orthogonality of 3D space). Importantly, directions like "up/down" and "left/right" are `relative` to the camera coordinate system, not absolute in the world coordinate system.
>
> On a broader note, our goal in this work is to highlight the potential of classic 3D tasks (like viewpoint tasks) in evaluating and enhancing MLLMs' spatial reasoning ability, rather than simply relying on a powerful base model trained on massive and diverse data. **We hope this insight inspires future work to more fine-grained task design and make more deliberate choices in data curation.**
>
> To further explore the ability, we create an up-down testset with `1000` examples. Each question has 3 options. So the random choice rate is `33.3%`. We test our Actial-7B and Qwen2.5-VL-7B-Instruct **without additional fine-tuning on up-down data.**
>
> And the accuracy of Actial-7B is `53.7%`, while the accuracy of Qwen2.5-VL-7B-Instruct is `21.2%`. Although training only on left-right data cannot perfectly solve the up-down task, the results still show a significant improvement over the base model. This suggests that future work should consider more comprehensive task designs.
>
>
> ## **Q2.2: Dataset diversity.**
>
> **Response:** We selected image pairs with angular differences between `±20°` and `±100°`. Given the full possible range of `±180°` in real-world scenarios, this range was chosen to balance task difficulty. Since both very small and very large viewpoint differences can be challenging even for humans. For example, in extreme cases where the offset angle is around 180 degrees, considering `possible errors`, either choosing a left rotation or a right rotation can be theoretically acceptable.
>
>
>
> Moreover, since MVImgNet consists of real-world images without fixed angular intervals, **the selected samples are naturally distributed across the ±20°–±100° range, which enhances the diversity of the dataset.**
>
> Finally, our dataset is generated `automatically` through a pipeline built on MVImgNet. If more challenging or diverse data are needed in the future, this can be easily achieved by adjusting the sampling parameters.
>
> ## **Q3: Object moving when camera is fixing.**
>
> **At first, the object moving task refers to the `Motion-Obj.` in **MMSI-Bench**, which Actial gains a `+2.6` improvement.**
>
> We are glad that you also noticed this point, and it is indeed one of the key insights from our work. The type of 3D reasoning you describe, where spatial changes are analyzed under a fixed camera setup, is more similar to a single-image spatial understanding task. Since the change between `multiple images` are occured more likely in the `timeline`, which is different with the `multi-view` images analysis.
>
> In the `SFT (K.L.)` phase, we fine-tuned the MLLM exclusively on multi-view viewpoint tasks. As typical with task-specific fine-tuning without diverse data, this can harm performance on out-of-domain tasks, such as the drop we observe on Orient. (3DSRBench) compared to the base model.
>
> Encouragingly, however, we find that even with fine-tuning on just this single task, the model shows `performance gains` on several out-of-domain single-image spatial reasoning tasks (the relative position questions like Height, Relation, and Depth), as seen in the improvement of "w/o G.E." over the base model. **This suggests that mastering viewpoint reasoning can generalize broadly and significantly enhance the model's overall spatial reasoning ability.**
>
> ## **Response to the weakness.**
>
> The computational cost of MLLM training is extremely high, particularly for `SFT+RL pipelines`. Given that many works in the field typically validate their methods on a single representative model, we adopt the widely used Qwen2.5-VL-7B-Instruct for our experiments. We thank the reviewer for the suggestion. However, due to constraints on resources and time (`the GRPO requires weeks of time`), we are unable to conduct additional experiments on other MLLMs during the rebuttal phase.

---

> > ### Comment · Reviewer_jwwJ · 2025-08-04
> >
> > The authors have mostly addressed my concerns. One follow-up question / clarification: I think the up-and-down viewpoint changing would also lead to different distribution of the object surface that's viewed from the camera. Consider the statue in Figure 2 in the paper, left-and-right means that the face of the statue is still within the same plane as the camera viewpoint. However, if considering up-and-down movements, the statue in the camera might only reveal the hair or the base, which results in different distribution of the image / object appearance in the image. I'm wondering the effect of such distribution difference, and whether there are data pairs inside the dataset that cover this scenario.

---

> > > ### Author Response · Authors · 2025-08-05
> > >
> > > We appreciate your acknowledgment of our response and thank you for your time and attention.
> > >
> > > We would first like to clarify one point: the current version of Viewpoint-100K only includes left-right translation and rotation queries. However, since the data is collected from real-world environments rather than being synthetically generated, adjacent image frames inevitably contain some degree of up-down offset, though the dominant motion remains horizontal translation and rotation.
> > >
> > > In response to your question, we will address it from two aspects:
> > >
> > > - **Would up-down movement introduce additional distributional changes in image/object appearance compared to left-right movement?**
> > >
> > >     `We appreciate this perspective, though our analysis indicates otherwise.`
> > >
> > >     The example you provided suggests that up-down movement might cause the camera to see only the hair or the base, while left-right movement reveals different facial aspects. However, this difference is not solely due to camera translation or rotation. It also depends on the object's 3D pose and its relative position and orientation with respect to the camera. For instance, if the statue were not placed vertically but instead laid horizontally on a table, then left-right movement would cause only the head or base to appear in the frame, while up-down movement would keep the head consistently visible.
> > >
> > > - **Does our dataset consist of a variety of scenarios, rather than only such head-visible examples?**
> > >
> > >     `We would answer yes, based on the following explanation.`
> > >
> > >     As previously mentioned, such appearance variations are jointly determined by the object’s 3D pose and the camera viewpoint. Our data is sourced from a large-scale, real-world multi-view dataset that captures diverse object configurations, such as upright, inverted, sideways, leaning against walls, stacked, or hanging. Therefore, even though we primarily select view transitions along the horizontal (left-right) direction, the real-world nature of the data ensures sufficient diversity and comprehensiveness in the final distribution.

---

> > > > ### Comment · Reviewer_jwwJ · 2025-08-06
> > > >
> > > > Thank you for your thorough response! All of my concerns have been addressed, and I will increase my score to 4. I agree that the example I provided also need to consider 3D object pose, which is covered in the real world diverse scenario.

---

> > > > > ### Author Response · Authors · 2025-08-06
> > > > >
> > > > > We deeply appreciate your encouraging comment. And we're glad the clarification was helpful.

---

### Official Review · Reviewer_ywk7 · 2025-07-02

**Clarity:** 3
**Significance:** 3
**Originality:** 3
**Rating:** 4
**Confidence:** 5

**Summary:**

This paper aims to bridge the gap between 2D vision-language models (VLMs) and 3D spatial reasoning. To this end, the authors introduce viewpoint learning, a new task to instill 3D awareness through reasoning over 2D multi-view images. Regarding this task, the authors propose Viewpoint-100K, a dataset comprising 100k object-centric multi-view images with corresponding QA pairs. To learn spatial reasoning ability, this paper employs Qwen-2.5-VL-7B-Instruct as the base model, and applies SFT on Viewpoint-100K followed by RL with GRPO on SAT dataset. The trained model demonstrates significant performance improvements on several spatial reasoning benchmarks, including 3DSRBench, CV-Bench, and BLINK.

**Questions:**

- I cannot get what it means and what the authors want to deliver in Figure 2. What does it mean by “Matched” or “Mismatched”?
- I would expect more results on Viewpoint-100K, including human performance and comparison between RL vs. SFT.
- There is little discussion about the quality of the pseudo CoT data (I guess Figure 4 does not refer to the collected CoT data). And if the quality is an issue, why still using a proportion for training instead of refining them in advance?
- Some “w/o K.L.” results in Table 1 outperform the full method. Does “w/o K.L.” mean the first SFT stage is removed? This should be clarified, and the results should be highlighted accordingly to aid interpretation.
- What is the observed impact of hybrid cold-start initialization using CoTs? Additional analysis or ablations would be helpful.
- Why name it Actial, any potential meaning or metaphor?

**Ethical Concerns:**

["NO or VERY MINOR ethics concerns only"]

**Final Justification:**

The authors addressed my concerns well. I agree that the paper exceeds the acceptance bar.

**Limitations:**

Yes

**Quality:**

2

**Strengths And Weaknesses:**

#### Strengths
- Good motivation. 3D spatial reasoning is a critical capability to be unlocked for VLMs. And this capability is particularly insufficient for current VLMs. This paper focuses on this significant problem and proposes viewpoint learning to mitigate the gap.
- Significant experimental results. The proposed approach greatly improves the baseline’s performance on spatial reasoning across multiple benchmarks, validating the effectiveness of the proposed dataset and viewpoint learning.
- Good clarity. The paper is well-written and logically structured, making it easy to follow. The authors clearly articulate the motivation, methodology, and results.

#### Weaknesses
- Collecting object-centric multi-view images is a good solution to cultivate 3D spatial reasoning skills. However, I think it may not be optimal to anchor the task on discriminating how the camera translates and rotates through texts. This task is quite difficult (even for humans), and learning can be challenging for models considering the task complexity and sparse supervision (only supervising short text answers). While how to properly formulate viewpoint learning remains an open question, I think it pertains to perceptual capability that should be embedded at the representation level rather than handled by language model reasoning. As such, casting it as a language task might not be the most effective route.
- The results and analyses on how to utilize Viewpoint-100K are not intuitive. The authors said that RL with GRPO leads to high KL divergence (i.e., a significant departure on the policy). I think this cannot support that RL is inadequate for overcoming the 2D-centric bias. And I think the improvement by applying direct SFT does not necessarily indicate a better foundamental spatial reasoning capability, considering the strong supervision on fitting the text. Anyway, this is not a convincing approach for knowledge injection from my perspective.

---

> ### Author Rebuttal · Authors · 2025-07-30
>
> ## **Q1: Explanation about Fig. 2.**
>
> **Response:** Thanks for all of your valuable comments, we will improve the clarity of the analysis in the manuscript.
>
> `Figure 2` is provided to help illustrate the difference between `2D continuity` and `3D consistency`. As stated in the Introduction of our paper:
>
> > "Our key concern is whether existing MLLMs, which are primarily trained on video data emphasizing 2D continuity, can achieve an understanding of spatial 3D consistency, as opposed to merely tracking continuous pixel-level evolution or correlated pixel mappings."
>
> In the figure, **"Matched"** represents the case with true 3D consistency. Under this condition, when images are transformed into the world coordinate system using accurate camera parameters, **pixels corresponding to the same 3D point across different views are correctly projected to the same location in 3D space**. This results in perfect alignment of the reconstructed object, with no misalignment or error.
>
> In contrast, **"Mismatched"** illustrates the scenario where only 2D continuity is preserved. When small, arbitrary `deformations` or `scaling` are applied to each frame, **adjacent frames appear visually smooth and continuous. However, the underlying 3D geometric consistency is broken.** In practice, such distortions resemble lens distortions, which require additional calibration to restore true 3D structure. In this case, corresponding points from different images `fail to align in 3D space`, resulting in misaligned projections or "ghosting" artifacts in the reconstructed object.
>
> When applied to MLLMs, this issue highlights **whether such models has the potential capability to map pixels carrying consistent information across different views back into the correct 3D space to perform crossing-view spatial reasoning**. And the Viewpoint Learning is the fundamental part of this issue.
>
> ## **Q2: More results on Viewpoint-100K testset.**
>
> **Response:** The testset of Viewpoint-100K includes `1000 examples`.
>
> - **Human performance:** We randomly sampled `100 examples` from testset and conducted a human evaluation with three annotators, achieving an average accuracy of `97.67%`. This high performance is mainly due to our deliberate design choice during data construction. To ensure reasonable task difficulty, we limited the angular difference between image pairs to between `±20` and `±100` degrees. This range helps maintain a challenging yet discriminable level of difficulty for the task.
>
> - **Qwen base:** The accuracy is `12.9%`.
>
> - **SFT:** The accuracy is `92.2%`.
>
> - **SFT + RL:** The accuracy is `81.4%`.
>
> ## **Q3: Quality of pseudo CoTs.**
>
> **Response:** Thank you for pointing out the missing discussion regarding the pseudo CoT data. The pseudo CoT examples are presented in `Figure 3 in the Appendix`. Figure 4 does not show collected CoT data; instead, it illustrates the current MLLM's performance on the viewpoint task.
>
> The lack of genuine spatial reasoning in current models is evident from the fact that even advanced models like Gemini perform at `random-guessing levels` on the BLINK viewpoint task. **This makes it unrealistic to rely on models to automatically generate high-quality CoT data. At the same time, manually correcting CoT data is prohibitively expensive.**
>
> To ensure the quality of the pseudo CoT data, we provide the correct answer to Gemini in advance to guarantee accuracy. We also supply a carefully curated, human-verified template that demonstrates correct spatial reasoning. Gemini then uses this answer and template as a reference to generate diverse CoT responses, effectively `imitating` the reasoning pattern. The prompt template used for generating the pseudo CoT can be found at the end of the Appendix.
>
> Furthermore, in our hybrid cold-start phase, the primary role of CoT data is to provide a formally correct reasoning template and maintain the model's instruction-following capability. This point is further discussed in our response to `Question 5`.
>
>
>
> ## **Q4: Question about "w/o K.L.".**
>
> **Response:** `"w/o K.L."` means the first SFT stage is removed. Thanks for the advise about clearer expression and visual prompts.
>
> In the `SFT (K.L.) phase`, we fine-tuned the MLLM exclusively on viewpoint tasks. Since SFT is `task-specific` and uses limited data diversity, it typically harms performance on tasks outside the fine-tuning distribution. This explains the observed performance drop on Orient. compared to the base model.
>
> However, we were pleasantly surprised to find that fine-tuning on viewpoint tasks led to performance `improvements on out-of-domain tasks`, such as Height, Depth, and Relation. This highlights the importance of viewpoint learning for enhancing the model's overall spatial ability.
>
> The fact that the "w/o K.L." variant outperforms our Actial on a few tasks can be attributed to the known negative impact of standard SFT on generalization. This is precisely why we introduce a second-stage GRPO. We apply RL across a broader set of tasks to enhance the generalization. Importantly, thanks to the knowledge injected during SFT, we observe that the model shifts from primarily relying on `2D visual cues` to engaging in `more genuine spatial reasoning`. It leads that the reasoning paths of the model after knowledge injection followed by GRPO (Actial) are `fundamentally different` from those in the direct GRPO approach (w/o K.L.).
>
>
> ## **Q5: Impact of hybrid cold-start initialization.**
>
> **Response:** The purpose is threefold.
>
> - First, to introduce `a structured template` for spatial reasoning. As shown in the thinking process in `Figure 5` of the paper, the model's reasoning pattern is shaped by the CoT examples from the cold-start phase.
>
> - Second, to maintain the model's `instruction-following capability`. As demonstrated in `Section 1  and Figure 1(a) of the Appendix`, the model struggles to learn the format reward without including CoT data during GRPO.
>
> - Third, since we cannot guarantee the correctness of the reasoning logic in the CoT templates, instead of using a separate cold-start phase between SFT and RL, we choose to incorporate the CoT data at a small ratio during our knowledge injection stage. This helps minimize the influence of potential template biases on the final performance.
>
> ## **Q6: The reason for naming "Actial".**
>
> **Response:** I'm glad our naming caught your attention. And **`"Actial"`** is the combination of **`[Ac]`tivate Spa`[tial]`** in our title, simultaneously simple and easy to read.
>
> ## **Additional comments about the RL and SFT for spatial reasoning.**
>
> We would like to share additional observations regarding the direct application of GRPO in Viewpoint Learning. Beyond the KL divergence issues, we found that the model's performance remains at a `random guessing level` throughout training. Despite multiple attempts, the training process consistently `collapsed` after a long time of training, characterized by a sudden drop in response length, after which the model began generating `meaningless or degenerate outputs`.
>
> We also examined the `CoT behavior` of a model trained directly on Viewpoint-100K using GRPO only. While we observed an initial `"aha moment"` in terms of increasing response length, further analysis revealed that the model still relied primarily on `2D visual cues` for reasoning, similar to what is shown in `Figure 4` of the main paper. This lack of genuine 3D spatial understanding was a key motivation for us to instead inject viewpoint knowledge via SFT, rather than relying solely on GRPO.
>
> ## **Additional discussions about the question settings.**
>
> In fact, from the initial stage to the current viewpoint learning approach presented in the paper, we `simplified the problem multiple times` to better fit the capabilities of current MLLMs.
>
> - Initially, we attempted to train the MLLM to directly regress the `relative rotation matrix` using RL, with a format reward and cosine similarity loss. However, the model failed completely to learn this precise regression task. This difficulty may stem from the inherent nature of LLMs (particularly the next-token prediction), which makes them less suited for fine-grained numerical regression. Additionally, rotation matrices are far removed from natural language semantics, which are the primary strength of LLMs.
>
> - To further probe the capabilities of MLLMs, we simplified the task by asking the model to predict relative rotation angles around a fixed axis. We tested both `direct angle regression` with error evaluation  and `multiple-choice selection`. Even with these simpler setups, the model struggled to learn effectively. This led us to question whether current MLLMs can truly develop accurate 3D spatial understanding and reasoning.
>
> - As a result, in the final version of our method, we adopted a more feasible approach using `relative directions`. Our results show that MLLMs, at least under current settings, can `handle basic 3D viewpoint tasks successfully`. And more difficult tasks like rotation and precise pose regression remain a challenging and important direction for future research.

---

> > ### Comment · Reviewer_ywk7 · 2025-08-05
> >
> > Appreciate the authors' response, which has addressed my concerns well. Glad to see the authors' additional discussion, and I agree that the evolution of their approach makes sense, as learning to describe viewpoint movement falls within the scope of what LLMs are relatively good at.

---

> > > ### Author Response · Authors · 2025-08-05
> > >
> > > Thank you for your thoughtful feedback and for recognizing the improvements. We truly appreciate your time and constructive comments.

---

### Official Review · Reviewer_AHFe · 2025-07-05

**Clarity:** 3
**Significance:** 2
**Originality:** 3
**Rating:** 4
**Confidence:** 3

**Summary:**

The paper tackles the lack of reliable 3-D spatial reasoning in multimodal LLMs. It frames Viewpoint Learning and introduces Viewpoint-100K dataset. Built on Qwen-VL-7B, the proposed two-stage pipeline: 1) supervised fine-tuning on Viewpoint-100K, mixing in 10 % Gemini-generated chains-of-thought. 2) reinforcement learning via GRPO on a synthetic SAT set. The resulting Actial-7B gains 12–14%  over the baseline on several spatial benchmarks.

**Questions:**

1. Could similar generalization be achieved by simply adding more diverse supervised data instead of RL? An ablation contrasting these options would help isolate the benefit of GRPO.

2. Please also clarify how the reward is defined in the GRPO stage: What components does it comprise (e.g., format reward and result reward) and how are they computed?

3. Why not compare against a variant that directly regresses relative poses (e.g., via cosine loss) rather than multiple-choice classification?

**Ethical Concerns:**

["NO or VERY MINOR ethics concerns only"]

**Final Justification:**

The authors addressed my concerns. I keep my score

**Quality:**

2

**Strengths And Weaknesses:**

[+] Well-motivated focus on 3-D consistency.
[+]  Simple two-stage recipe with complementary gains, supported by ablations.
[+]  Insightful error analysis illustrating shift from 2-D shortcuts to 3-D reasoning.

[-]  Performance gap to proprietary SOTA remains; no significance tests.
[-]  Lack of ablation study on the RL training. Contrast RL stage with simply scaling supervised data.

---

> ### Author Rebuttal · Authors · 2025-07-31
>
> ## **Q1: SFT ablation and performance against SOTA.**
>
> **Response:** Thank you for pointing out the absence of this ablation study. It helps improve the quality of our work.
>
> We conducted this additional ablation experiment by mixing Viewpoint-100K, pseudo CoTs, and SAT. We use SFT `(without GRPO)` to fine-tune Qwen2.5-VL-7B-Instruct using the same training parameters as in the previous experiments. We trained for a total of 2,000 steps (approximately 1.5 epochs).
>
> The evaluation is conducted on the `more comprehensive and challenging` **MMSI-Bench**[1]. Due to our long CoT template length, we set the max_tokens to 32K.
>
>
> | MMSI-Bench | Param | Cam.-Cam. | Obj.-Obj. | Reg.-Reg. | Cam.-Obj. | Obj.-Reg. | Cam.-Reg. | Meas. | Appr. | Motion-Cam. | Motion-Obj. | MSR | Avg. |
> | -- | -- | -- | -- | -- | -- | -- | -- | -- | -- | -- | -- | -- | -- |
> | GPT-4o | - | **34.4** | 24.5 | 23.5 | 19.8 | **37.6** | 27.7 | 32.8 | **31.8** | **35.1** | **36.8** | **30.8** | 30.3 |
> | Qwen2.5-VL-72B | 72B |25.8 | **34.0** | 34.6 | 23.3 | 34.1 | 36.1 | **45.3** | 27.3 | 27.0 | 30.3 | 27.3 | **30.7** |
> | InternVL2.5-78B | 78B | 23.7 | 22.3 | **39.5** | 29.1 | 31.8 | **42.2** | 35.9 | 19.7 | 17.6 | 26.3 | 27.3 | 28.5 |
> | Qwen2.5-VL-7B-Instruct | 7B | 23.7 | 24.5 | 19.8 | 25.6 | 32.9 | 33.7 | 42.2 | 24.2 | 18.9 | 30.3 | 23.2 | 26.5 |
> | **Actial-7B(Ours)** | 7B | 29.0 `(+5.3)` | 31.9 `(+7.4)` | 28.4 `(+8.6)` | **41.9** `(+16.3)` |28.2 (-4.7) | 33.7 | 31.3 (-10.9) | 22.7 (-1.5) | 27.0 `(+8.1)` | 32.9 `(+2.6)` | 20.7 (-2.5) | 28.9 `(+2.4)` |
> | **Ablation Model** | 7B | 19.4 `(-9.6)` | 29.8 `(-2.1)` | 27.2 `(-1.2)` | 31.4 `(-10.5)` | 30.6 (+2.4) | 34.9 (+1.2) | 29.7 `(-1.6)` | 19.7 `(-3.0)` |  25.7 `(-1.3)` |  25.0 `(-7.9)` | 26.3 (+5.6) | 27.2 `(-2.6)` |
>
>
>
> **Actial** achieves comprehensive improvements in most subtasks when compared to the **Ablation Model**. These results demonstrate the effectiveness of our two-stage training framework and highlight the significant performance gains achieved through GRPO in out-of distribution tasks.
>
>
> Moreover, our model, with `only 7B parameters`, achieves performance comparable to that of larger models and GPT-4o, and even outperforms them on certain tasks.
>
> The significant difference on the `MSR task` is due to Actial generating excessively `long reasoning chains` when handling multi-step inference, causing the output to `be truncated` before the correct answer is reached. This prevents the model from producing complete and accurate responses. Addressing this issue by optimizing reasoning efficiency and managing output length will be an important direction for future work.
>
> [1] Yang et al. 2025. MMSI-Bench: A Benchmark for Multi-Image Spatial Intelligence
>
> ## **Q2: Reward details.**
>
> **Response:** Our reward score consists of two components.
>
> - The first part is the `format reward`, which encourages the model to use the correct reasoning format (specifically, placing thoughts within `\<think>` and `\</think>` tags).
> - The second part is the `result reward`, which evaluates whether the model selects the correct answer from the multiple choices. We extract the final answer using regular expressions and compare it to the ground truth.
> - Both components are equally weighted, with a ratio of `1:1`.
>
> ## **Q3: Pose regression.**
>
> **Response:** We are pleased that you also noticed the issue of precise pose regression using MLLMs. We explored similar ideas in our early experiments. In fact, from the initial stage to the current viewpoint learning approach presented in the paper, we `simplified the problem multiple times` to better fit the capabilities of current MLLMs.
>
> - Initially, we attempted to train the MLLM to directly regress the `relative rotation matrix` using RL, with a format reward and cosine similarity loss. However, the model failed completely to learn this precise regression task. This difficulty may stem from the inherent nature of LLMs (particularly the next-token prediction), which makes them less suited for fine-grained numerical regression. Additionally, rotation matrices are far removed from natural language semantics, which are the primary strength of LLMs.
>
> - To further probe the capabilities of MLLMs, we simplified the task by asking the model to predict relative rotation angles around a fixed axis. We tested both `direct angle regression` with error evaluation  and `multiple-choice selection`. Even with these simpler setups, the model struggled to learn effectively. This led us to question whether current MLLMs can truly develop accurate 3D spatial understanding and reasoning.
>
> - As a result, in the final version of our method, we adopted a more feasible approach using `relative directions`. Our results show that MLLMs, at least under current settings, can `handle basic 3D viewpoint tasks successfully`. And more difficult tasks like rotation and precise pose regression remain a challenging and important direction for future research.

---

### Decision · Program_Chairs · 2025-09-17

**Decision:**

Accept (poster)

**Comment:**

This paper was reviewed by four experts who all recommended acceptance of this paper.  The major contribution of this paper is a dataset to evaluate a MLLM's performance on understanding different viewpoints of the same object.  The authors also provide an approach for finetuning MLLMs on their task, which the rebuttal points out outperforms some simpler competing finetuning methods.  To help demonstrate the benefits of their dataset, the authors use a model trained on their dataset to evaluate on external spatial reasoning benchmarks.  Overall, this helps highlight how data constructed using their approach can be beneficial to a wider range of spatial reasoning datasets and tasks.  One somewhat counterintuitive result was lower performance on the Orientation questions of the 3DSRBench, which would seem closely related, but the authors argued this stemmed from the differences in the questions- where 3DSRBench did not include a reference image.  This highlights a potential area of weakness and limited impact of data trained using this dataset, calling into some question as to the true source of the gains (i.e., does the proposed dataset improve performance on these other datasets due to training on object-focused data, focusing the MLLM weights to think about the objects rather than other aspects like color, or background information).  However, these questions can easily be argued as simple conjecture, where the current dataset provides a reasonable benchmark with some demonstrated benefits.  As such, the ACs find insufficient justification for overturning the reviewer's recommendations, and the authors are encouraged to consider reviewer comments carefully when preparing their camera ready.